

# Investigation of NO₂ vertical distribution using two DOAS retrievals for GOME-2A measurements in the UV and vis spectral range

Lisa K. Behrens[1], Andreas Hilboll[1,2], Andreas Richter[1], Enno Peters[1], Henk Eskes[3], and John P. Burrows[1]

[1]Institute of Environmental Physics (IUP-UB), University of Bremen, Bremen, Germany
[2]MARUM - Center for Marine Environmental Sciences, University of Bremen, Bremen, Germany
[3]Royal Netherlands Meteorological Institute (KNMI), De Bilt, the Netherlands

*Correspondence to:* Lisa K. Behrens (lbehrens@iup.physik.uni-bremen.de)

**Abstract.** In this study, we present a novel NO₂ DOAS retrieval in the ultraviolet (UV) spectral range for satellite observations from the Global Ozone Monitoring Instrument 2 on board EUMETSAT's MetOp-A (GOME-2A) satellite. We compare the results to those from an established NO₂ retrieval in the visible (vis) spectral range from the same instrument and infer information about the NO₂ vertical profile shape in the troposphere.

As expected, radiative transfer calculations for satellite geometries show that the sensitivity close to the ground is higher in the vis than in the UV spectral range. Consequently, NO₂ slant column densities (SCDs) in the vis are usually higher than in the UV, if the NO₂ is close to the surface. Therefore, these differences in NO₂ SCDs between the two spectral ranges contain information on the vertical distribution of NO₂ in the troposphere. We combine these results with radiative transfer calculations and simulated NO₂ fields from the TM5 chemistry transport model
to evaluate the simulated NO₂ vertical distribution.

We investigate regions representative for both anthropogenic and biomass burning NO₂ pollution. Anthropogenic air pollution is mostly located in the boundary layer close to the surface, which is reflected by the large differences between UV and vis SCDs of ∼60 %. Biomass burning NO₂ in contrast is often uplifted into elevated layers above the boundary layer. This is best seen in tropical Africa south of the equator, where the biomass burning NO₂ is well
observed in the UV, and the difference between the two spectral ranges is only ∼36 %. In tropical Africa north of the equator, however, the biomass burning NO₂ is located closer to the ground, reducing its visibility.

While not enabling a full retrieval of the vertical NO₂ profile shape in the troposphere, our results can help to constrain the vertical profile of NO₂ in the lower troposphere and, when analyzed together with simulated NO₂ fields, can help interpret the model output.



## 1   Introduction

Nitrogen dioxide ($NO_2$) is an important indicator for natural phenomena and anthropogenic air pollution, as it is produced in the troposphere by, e.g., biomass burning or combustion of fossil fuels (Lee et al., 1997). Additionally, $NO_2$ is produced by lightning (Lee et al., 1997; Beirle et al., 2004) and microbiological soil emissions (Williams et al.,

1992; Bertram et al., 2005). The relevance of the individual $NO_2$ sources depends on the region of the Earth (van der A et al., 2008). Biomass burning is important in equatorial regions like central Africa, whereas anthropogenic air pollution is mostly important in industrialised areas like China or Europe. The distribution of $NO_2$ is of major interest because it is harmful to human health, adds to local radiative forcing, catalyses surface ozone production during summer smog, and causes acid rain (Finlayson-Pitts and Pitts, 1999).

Using spectrometers, solar radiation scattered upwelling at the top of the Earth's atmosphere can be measured and the amount of trace gases inverted mathematically from the depth of molecular absorption bands. Such measurements have been made from satellite (e.g., Burrows et al., 1999), airborne (e.g., Heue et al., 2005; Wang et al., 2005), and ground-based (Noxon, 1975) platforms.

Since hyperspectral satellite observations began in the mid-1990s (Burrows et al., 1999), the horizontal distribu-
tion of the column amounts of many trace gases is well known, e.g., for $NO_2$. Using sun-synchronous orbits, a nearly daily global coverage at similar local times can be obtained. The global coverage as well as the spatial resolution of the measurements depends on the instrument (see for example: Callies et al., 2000). However, there is only limited knowledge of the vertical distribution from satellite data. In contrast, ground based Multi AXis Differential Optical Absorption Spectroscopy (MAX-DOAS) measurements have high temporal resolution and can provide vertical pro-
files of trace gases up to several kilometres altitude (Wittrock et al., 2004), but can only measure at one particular location and are thus limited in resolving horizontal gradients.

Irrespective of the instrument's viewing geometry, the measured spectra can be analysed using the Differential Optical Absorption Spectroscopy (DOAS; Platt and Stutz, 2008; Burrows et al., 2011) technique which is a well established method based on Lambert Beer's law discribing the spectral reduction of the initial intensity of light
due to absorption. The main result is the integrated concentration of trace gases $\rho(s)$ along the effective light path $s$ which is called the total slant column density (SCD; Platt and Stutz, 2008; Burrows et al., 2011):

$$SCD = \int \rho(s)ds. \tag{1}$$

As Rayleigh scattering in the atmosphere depends strongly on wavelength ($\lambda$; the scattering cross-section is proportional to $\lambda^{-4}$), sun light having longer wavelengths has a larger penetration depth in the atmosphere compared
to shorter wavelengths. This results in a wavelength-dependency of the SCDs (Burrows et al., 2011). Consequently, if $NO_2$ is located close to the ground, SCDs retrieved from satellite measurements at shorter wavelengths are in theory smaller than SCDs retrieved from longer wavelengths. In consideration of this fact, knowledge of the vertical distribution of $NO_2$ can be gained by combining measurements at different wavelengths. The idea of using the penetration depth in the UV to determine vertical profiles of ozone was first proposed by Singer and Wentworth





(1957). The use of the temperature dependence of the Huggins absorption bands coupled with penetration depth was proposed to retrieve information about the vertical profile of ozone in the troposphere (Chance et al., 1997).

In the presence of clouds, the described behaviour changes. For cloudy scenes only smaller differences are expected. Due to the clouds the sensitivity towards the surface is decreased, and therefore, the $NO_2$ below the clouds is partly invisible for the instrument. For the $NO_2$ above the clouds, the sensitivity is similar or partly increased for the UV spectral range (Burrows et al., 2011). In previous studies, clouds at different altitudes are used to obtain information about $NO_2$ profiles (Choi et al., 2014; Belmonte Rivas et al., 2015).

The vertical column densities (VCDs) are the integral of the trace gas concentration from the surface to the top of the atmosphere along the altitude $z$ (Platt and Stutz, 2008; Burrows et al., 2011):

$$VCD = \int \rho(z) dz. \tag{2}$$

They can be calculated using air mass factors (AMFs), which are defined as the ratio of SCDs and VCDs (Platt and Stutz, 2008; Burrows et al., 2011), and are an indicator of the measurement sensitivity or the length of the light path within the $NO_2$ layer:

$$AMF(\lambda) = \frac{SCD(\lambda)}{VCD}. \tag{3}$$

AMFs are calculated by radiative transfer models, which take into account the viewing geometry and environmental effects (Platt and Stutz, 2008). The sensitivity of the measurement to an absorber varies with altitude; this is expressed by the so-called box air mass factor (BAMF; Burrows et al., 2011), which is defined as $BAMF_i = SCD_i/VCD_i$ for an atmospheric layer $i$. For wavelengths in the UV, the BAMF in layers close to the ground is considerably smaller than for vis wavelengths. In general, BAMFs for longer wavelengths have a smaller dependency on altitude compared with BAMFs for shorter wavelengths (Burrows et al., 2011). Furthermore, the surface spectral reflectance (SSR) depends on the wavelength, and therefore, the SCDs are influenced by the SSR (Burrows et al., 2011). For smaller SSR, the UV shows a stronger decrease and for larger SSR the UV shows stronger increase towards the surface. Therefore, the SSR can strengthen the effect of the Rayleigh scattering. Generally, the SSR is for the UV smaller than for the vis spectral range. Additionally, aerosols influence the measurements and also the visibility of $NO_2$ is influenced by the presence of aerosols (Burrows et al., 2011). Depending on the type and the optical thickness of aerosols the influences on the measurement differs.

Another parameter needed for the AMF calculation is an a-priori $NO_2$ profile (Burrows et al., 2011), as the total AMF is calculated as the average of the BAMFs of all atmospheric layers, weighted by the absorber concentration. The retrieved VCDs therefore, depend on the a-priori $NO_2$ profiles, and differences between the a-priori and actual $NO_2$ profiles can introduce systematic errors in the VCDs. As in principle, the final VCDs should not depend on the wavelength, analysing the differences in the VCDs retrieved from different wavelength regions can provide information about the accuracy of the used a-priori $NO_2$ profiles.

Typically, $NO_2$ DOAS fitting windows used in satellite data analysis are in the vis spectral range (see for example: Richter et al., 2011 and van Geffen et al., 2015). The only satellite $NO_2$ retrieval sofar using UV wavelengths has





been developed for the Ozone Mapper and Profiler Suite (OMPS) on board of the Suomi National Polar-orbiting Partnership (SNPP) satellite, employing a DOAS-like method to derive $NO_2$ VCDs (Yang et al., 2014). Compared to VCDs retrieved in the vis spectrum with DOAS from the Ozone Monitoring Instrument (OMI) on board the Aura satellite, Yang et al. found good agreement between the two retrievals.

In the present study, we present a DOAS retrieval for $NO_2$ in the UV spectral range and use the results and compare this with a DOAS retrieval in the vis spectral range, which is our standard approach, to investigate the vertical sensitivity of satellite-based $NO_2$ measurements. In Sect. 2, the $NO_2$ DOAS retrieval in the UV spectral range will be introduced for measurements from the Global Ozone Monitoring Experiment 2 (GOME-2) on board MetOp-A. The UV $NO_2$ retrieval settings will be discussed in detail as well and the dataset from the vis spectral

range will be presented. In Sect. 3, results for the two $NO_2$ retrievals in the UV and vis spectral ranges will be compared, and their implications for the $NO_2$ vertical distribution will be discussed. The manuscript ends with conclusions and a summary in Sect. 4.

## 2    Datasets and methods

### 2.1    The GOME-2A instrument

Different types of remote sensing instrumentation on satellite platforms yield spectral radiance at the top of the atmosphere, which yield the amounts and distribution of $NO_2$ in the Earth's atmosphere. Here, we use the GOME-2 instrument (Callies et al., 2000) flies on board the MetOp-A satellite (hereafter GOME-2A), which has a swath width of 1920 km resulting in nearly global coverage every day. MetOp-A was launched in October 2006 into a sun-synchronous orbit with an equator crossing time of 09:30 local time (LT) in descending node. The GOME-2A

instrument is a nadir viewing instrument with a ground pixel size of $80 \times 40\,km^2$ (Callies et al., 2000). In July 2013, the ground pixel size was reduced to $40 \times 40\,km^2$, when the identical GOME-2 instrument on board MetOp-B (hereafter GOME-2B), launched in September 2012, took over operational measurements (Munro et al., 2016). The spectrometer is separated into four channels covering wavelengths from 240 to 790 nm (Callies et al., 2000). Channel 2 provides data in the UV spectral range from 311 to 403 nm with a spectral resolution of 0.26 to 0.28 nm, while

channel 3 provides data in the vis spectral range from 401 to 600 nm with a spectral resolution of 0.44 to 0.53 nm. These two channels are used for the following analysis.

### 2.2    $NO_2$ DOAS retrieval in the UV spectral range

For this study, we developed a $NO_2$ DOAS retrieval for the GOME-2A instrument in the UV which uses a fitting window between 342 and 361.5 nm, and a polynomial degree of four. In this retrieval, one $NO_2$ cross section at 243 K

measured with the GOME-2A instrument (Gür et al., 2005) as well as two $O_3$ cross sections are used (at 223 K and 243 K, Serdyuchenko et al., 2014) with an additional $I_0$ correction of $10^{20}\,molec\,cm^{-2}$ (Platt et al., 1997; Richter,



1997). Additionally, cross sections for $O_4$ (Greenblatt et al., 1990), BrO (Wilmouth et al., 1999), HCHO (Meller and Moortgat, 2000), the Ring effect (calculated with SCIATRAN, Rozanov et al., 2014) and the instrumental cross section Zeta (EUMETSAT, 2011) are included in the fitting procedure (see Tab. 1).

[Table 1 about here.]

5    These settings are the result of a number of careful sensitivity tests. Among many different wavelength windows we tested, the selected window from $342 - 361.5\,\mathrm{nm}$ provided the best results. The instrumental correction function Eta (EUMETSAT, 2011) only had minor influence on the results when included in the fit; we therefore chose not to include it in the fitting procedure to keep the number of fit parameters small. Including additional $O_3$ cross sections to correct the $O_3$ non-linearity (Puķīte et al., 2010) improved the fit in polar latitudes, where solar angles and thus 10    $O_3$ absorptions are large. However, this also introduced an offset in the data, and as we are mainly interested in polluted and not in polar areas, we chose not to use it here. Finally, we investigated the effect of using a daily Earth reflectance spectrum as reference. However, the differences between the two reference spectra was only minor so that we use the solar reference in order to preserve consistency with the fit in the vis wavelength region (see Sect. 2.3).

### 2.3 NO₂ DOAS retrieval in the visible spectral range

15 Due to the considerably stronger absorption lines in the vis compared to the UV spectral range (Fig. 1), $NO_2$ DOAS retrievals normally use measurements in the vis spectral range, see for example Richter et al. (2011) or van Geffen et al. (2015). Here for the vis spectral range, a retrieval from $425 - 450\,\mathrm{nm}$ with a polynomial degree of four is used (introduced in Burrows et al., 1999 for the GOME instrument, applied to GOME-2A measurements in Richter et al., 2011). The fit settings are summerized in Tab. 1. Cross sections for $NO_2$ at $243\,\mathrm{K}$ and $O_3$ at $223\,\mathrm{K}$ are used for this 20    retrieval. Additional cross section are $O_4$, $H_2O$, and the Ring effect.

[Figure 1 about here.]

### 2.4 Comparison of the NO₂ DOAS retrievals in the UV and visible spectral range

In the UV spectral range, the differential $NO_2$ absorption lines are weaker than in the vis spectral range. In Fig. 1 the $NO_2$ cross section (Gür et al., 2005) at $243\,\mathrm{K}$ measured with the GOME-2A instrument is shown. Coloured shaded 25    areas indicate the fitting windows in the UV (green) and vis (blue) spectral range. The wavelength dependency of the differential absorption strength is clearly observed. Furthermore, the measurement sensitivity for $NO_2$ decreases towards the surface (Fig. 2). In combination with reduced intensity from the sun, this leads to a considerably higher noise level in the $NO_2$ differential optical depths in the UV compared to the vis. To illustrate this, Fig. 3 shows an example for the UV (a) and vis (b) $NO_2$ fit results for one measurement above Teheran ($35.38°\,\mathrm{N}$, $51.47°\,\mathrm{E}$) on 22 30    January 2008. The Figure shows the $NO_2$ reference (soild lines) and reference plus the unexplained part of the $NO_2$ signal, called *residual* (dashed lines). As expected, the retrieved $NO_2$ SCD is often smaller in the UV spectral range



compared to the vis (for the measurement shown in Fig. 3, the retrieved SCDs are $6.31 \times 10^{16}$ molec cm$^{-2}$ in the UV and $9.33 \times 10^{16}$ molec cm$^{-2}$ in the vis spectral range).

[Figure 2 about here.]

[Figure 3 about here.]

5    Due to the noisier differential optical depth signal, the UV NO$_2$ fit has larger uncertainty compared to the vis NO$_2$ fit. The UV NO$_2$ fit of the example has a random error of 4.3 %, whereas the vis NO$_2$ fit has a smaller error of 0.8 %. This is visible in Fig. 4 which shows the distribution of total NO$_2$ VCDs over a presumably clean area (equatorial Pacific from 5° S to 5° N and from 150° E to 210° E). Here, a stratospheric AMF was applied to calculate the NO$_2$ VCDs, as the common assumption is that no or very little tropospheric NO$_2$ is present in this area (Richter and Burrows, 2002; Martin et al., 2002; Peters et al., 2012). Due to the lower fit quality in the UV, the NO$_2$ columns retrieved there have a larger standard deviation of $7.4 \times 10^{14}$ molec cm$^{-2}$ compared to the vis with a standard deviation of $2.1 \times 10^{14}$ molec cm$^{-2}$.

[Figure 4 about here.]

Generally, both NO$_2$ retrievals lead to similar SCD patterns. In Fig. 5 NO$_2$ SCDs for the UV and vis spectral range are shown for one orbit passing above East China on 03 January 2008. The larger uncertainty of the UV NO$_2$ retrieval is reflected in this figure by the larger spread of the slant columns and the existence of unphysical negative values. Nevertheless, the NO$_2$ columns in the UV and vis spectral range are comparable for background regions and show a similar dependency on latitude. Over highly polluted regions, as for example China (25° N $-$ 50° N), the vis NO$_2$ values are larger compared to the NO$_2$ values in the UV spectral range.

20                            [Figure 5 about here.]

## 2.5   Removing stratospheric and upper atmospheric NO$_2$

For the comparison of tropospheric SCDs and VCDs, a correction for the impact of stratospheric NO$_2$ is needed. For the SCDs, we use the "reference sector method" (Richter and Burrows, 2002; Martin et al., 2002), in which a monthly average of SCDs measured over a presumably clean area above the Pacific (180° E to 210° E) is subtracted from all measurements per latitude band. The underlying assumption for this simple correction method is that no NO$_2$ is present over the reference sector in the troposphere, and that stratospheric NO$_2$ is zonally homogeneously distributed (Richter and Burrows, 2002; Martin et al., 2002). However, as this assumption is not always valid (Richter and Burrows, 2002; Boersma et al., 2004; Hilboll et al., 2013b), the reference sector method leads to areas with negative tropospheric NO$_2$ values, which are related to zonal inhomogeneities in stratospheric NO$_2$, for example close to the polar vortex (Dirksen et al., 2011). For this correction method, the same cloud screening as for the data





selection is used (see Sect. 2.7). As the sensitivity to the stratosphere is nearly similar for both spectral ranges (see Sect. 2.6), this correction can be used.

For the VCDs, we use a more sophisticated correction method, in which stratospheric VCDs from the Bremen-3D chemistry transfer model (B3dCTM, see Hilboll et al., 2013b and references therein) are used for the stratospheric correction. Therefore, no negative values are expected for the VCDs. A Lambertian surface is assumed and the Lambert-equivalent reflecty is taken as the SSR. For the AMF, a cloud correction is applied using the independent pixel approximation and cloud radiance fractions derived from the Fast REtrieval Scheme for Clouds from the Oxygen A-band (FRESCO+, version 6; Wang et al., 2008) dataset. Furthermore, the temperature dependency of the $NO_2$ cross section is included by using a linear approach as suggested in Boersma et al., 2004.

## 2.6 Radiative transfer simulations

The vertical sensitivity of the measurements to $NO_2$ has been investigated using box air mass factors simulated for both the UV ($352\,nm$) and the vis ($438\,nm$) fitting windows by the radiative transfer model SCIATRAN (version 3.6.5, see Rozanov et al., 2014). To calculate the radiance SCIATRAN requires knowledge about the measurement scenario, e.g., viewing geometry, solar position, atmospheric absorbers, spectral surface reflectance, surface elevation, aerosols and clouds as input parameters. Here, SSR and surface elevation have been taken from the OMLER_V003 (5-year climatology $2005-2009$; wavelength: $354\,nm$ and $442\,nm$; Kleipool et al., 2008) and GMTED2010 (Danielson and Gesch, 2011) datasets, respectively. Exemplarily, results for 30°, 50° and 70° solar zenith angle (SZA) and SSR of 0.04 (UV) and 0.06 (vis) are shown in Fig. 2. The SSR differs between the two spectral ranges; the two SSR values are chosen to be representative of soil. For 50° SZA, the sensitivity at the surface in the vis is about three to four times larger than in the UV. The sensitivity difference between the two wavelength regions decreases with increasing altitude, until at about $9\,km$, they are identical. Above $9\,km$ the UV shows a slightly (up to $4\,\%$) higher sensitivity to $NO_2$ compared to the vis spectral range. With higher altitude the BAMFs converge, up to at around $30\,km$, there are no significant differences between the UV and the vis spectral range (not shown). As shown in Fig. 2, the point of identical sensitivity increases slowly with increasing SZA which also influences the sensitivity in the atmosphere below the point of identical visibility. However, close to the ground, the differences between to two wavelength ranges stay nearly constant.

In order to calculate the effective AMF for one measurement scene, the BAMF profile needs to be combined with an a-priori vertical profile. This a-priori profile has a strong impact on the AMFs, as they provide the basis for the effective visibility of the $NO_2$ in the measurement scene. In this study, we used vertical $NO_2$ profiles from the TM5-MP model, described in Williams et al., 2017. For the European project QA4ECV (http://www.qa4ecv.eu) a 15 year run ($2002-2016$) was performed with the TM5-MP model version of July 2016. The model was run with a resolution of $1° \times 1°$ on 34 levels, using ERA-Interim reanalysis meteorology from ECMWF. Output was provided for HCHO, $SO_2$, $NO_2$ and $O_3$ with a temporal sampling every 2 hours. This dataset has been used within the QA4ECV project as a-priori for the retrieval of e.g. HCHO, and for other projects such as the one described here.





In particular, biomass burning emissions are taken from the monthly estimates provided by the GFEDv3 inventory (van der Werf et al., 2010) and latitude-dependent injection heights and a tropical burning cycle are implemented following Huijnen et al., 2010. See Williams et al., 2017 for other model details.

## 2.7 Data selection and post-processing

Both the UV and vis dataset were gridded to a $0.25° \times 0.25°$ grid and monthly means were calculated. Because of the fit quality and the reduced intensity in the UV spectral range, only measurements at solar zenith angles smaller than 70° are included in the following analyses. Only measurements with a geometric cloud fraction smaller than 0.2 are included, unless otherwise noted. Cloud filtering was performed using the FRESCO+ (version 6) dataset (Wang et al., 2008). Furthermore, only fits with a $\chi^2$ (describing the fit residual) smaller than 0.005 and 0.001 for the UV and vis spectral range, respectively, were used. While for the first years of the GOME-2A measurements, a consistent $\chi^2$-limit could be used for both spectral ranges, instrumental degradation severely impacts the instrument's performance in later years (Dikty and Richter, 2011). Since channel 2 is more strongly affected by this degradation than channel 3, a larger $\chi^2$-limit for the UV spectral range is needed for a statistically meaningful comparison.

## 3 Results and Discussion

### 3.1 Spatial distribution of NO$_2$ slant columns

Slant column densities (SCDs) are retrieved from the DOAS fit; they do not depend on any a-priori assumptions on the state of the atmosphere. Therefore, a comparison of the spatial distribution of the SCDs from the two UV and vis fit windows provides a first opportunity to assess the NO$_2$. However, the SCDs depend on the measurement geometry, SSR and wavelength. Consequently, maps of SCDs show not only the altitude dependency, but also changes of the AMF. For naturally polluted areas, the AMFs for the vis and the UV spectral ranges differ by a factor of $\sim 0.9 - 1.1$ (not shown) whereas the changes in SCDs differ between a factor of $\sim 1.1$ (northern hemispheric, NH, summer in Africa north of the equator; ANE) and $\sim 2$ (see for example Fig. 9). For anthropogenically polluted areas, the AMFs differ by a factor of $\sim 0.9 - 1.3$ whereas the SCDs differ by a factor of $\sim 2 - 3$ or more. Consequently, the part of the SCD differences can be explained by the different AMFs, but other factors like injection height and relative vertical distribution have to contribute as well.

Figure 6 shows monthly averages of tropospheric NO$_2$ SCDs for January (upper panels) and July (lower panels) 2008. Tropospheric NO$_2$ SCDs are shown for both, UV (a, c) and vis (b, d) fits. In both spectral ranges, similar spatial patterns are found and anthropogenic as well as natural air pollution can be detected. Especially over the respective winter hemisphere, anthropogenic source regions are clearly observed. In January, for example, the highest NO$_2$ columns are located over China (a, b), whereas in July, there are high NO$_2$ values over the Highveld Plateau region in South Africa (HPSA; c, d). In addition, biomass burning regions are also clearly visible in the data. For




instance, in July over Africa south of the equator (ASE), enhanced values are detected in both spectral ranges, consistent with van der A et al. (2008) and Schreier et al. (2014), who found a similar seasonal pattern. Finally, artefacts originating from the simplified stratospheric correction are observed in Fig. 6 (a, b), when enhanced $NO_2$ values over the Pacific reference area lead to large areas with too low $NO_2$ values in the subtropics of the northern

hemisphere (see Sect. 2.5) in both spectral ranges.

[Figure 6 about here.]

In order to better compare the two $NO_2$ retrievals, Fig. 7 shows the absolute differences between these results. Over highly polluted areas, $NO_2$ values in the UV spectral range are generally lower than in the vis spectral range (see Sect. 2.4). These differences are introduced by the wavelength dependency of the penetration depth (see Sect. 1),

which leads to better visibility of lower tropospheric pollution to the vis retrieval. Larger differences between the two $NO_2$ retrievals can be found in the respective winter hemisphere, e.g., in January over China and the east coast of the USA or in July over HPSA (see also Fig. 6). These large differences are related to first, the larger tropospheric $NO_2$ SCDs, and second, the larger SZA (see Fig. 2). In addition, large differences are observed in the western South Atlantic, where the retrieval noise in both spectral ranges is strongly enhanced due to the South Atlantic Anomaly

(Richter et al., 2011).

[Figure 7 about here.]

To further analyse the differences between the two spectral windows, we use the ratio of UV to vis tropospheric $NO_2$ SCDs which is a first approximation of an altitude dependency; Fig. 8 shows these for January (a) and July 2008 (b). To visualise the ratio of the two $NO_2$ retrievals, a threshold value was used to filter out pixels where

a too low denominator (due to noise) would lead to unrealistically high ratios. The threshold was defined as a smoothed latitude-dependent mean over the reference sector area. The Pacific threshold is one standard deviation of the gridded $NO_2$ values for both retrievals. To smooth the latitude-depend threshold, a 5°-running-mean is used.

Highly polluted areas are clearly visible, with ratios as low as $\sim 0.2$. As before, differences between the two retrievals are larger in the respective winter hemisphere, showing as lower ratios. The most obvious example is East China in

January, where large areas with very low ratios can be detected. However, all strong anthropogenic source regions, e.g., India, northern America, HPSA, and the Middle East, but also individual large cities like Madrid, Moscow, and Mexico City, show low ratios between the two retrievals. Here, it should be noted that due to the rather strict selection criterion on the SZA (see Sect. 2.7), no data are available for the strong source regions in central Europe in January.

[Figure 8 about here.]

In addition to anthropogenic air pollution close to the ground, $NO_2$ pollution from biomass burning can be detected in the retrieval ratios, e.g., in ANE and ASE (Fig. 8). In July (southern hemispheric, SH, winter), values of $\sim 0.6$



are found for the UV / vis ratio over the ASE region. These low values (as well as the absence of a signal in January) correspond to the seasonal pattern of biomass burning in this region.

In the ANE region, the vis fit results show clearly enhanced $NO_2$ SCDs in January (NH winter, coinciding with the annual biomass burning peak in this region) but not in July (see Fig. 6). However, no significant differences

between the two months are observed in the UV fit results, where slightly enhanced values can be seen throughout the year (not shown). These findings will be further discussed in Sect. 3.2.

North of the ANE region at approx. $10° - 20°$ N, SCD ratios of $\sim 0.7$ can be found in July (see Fig. 8). This is an indicator for $NO_2$ enhancement most probably related to soil emissions (Jaeglé et al., 2004; Zörner et al., 2016). While this $NO_2$ is more clearly observed in the vis spectral region, also the UV results show enhanced SCDs (see

Fig. 6). As shown by Delon et al. (2008) and Stewart et al. (2008), $NO_x$ ($NO_x = NO + NO_2$) from soil emissions is usually well mixed and thus not only located close to the ground, but also in elevated layers, enhancing visibility in the UV.

Finally, Fig. 8 shows SCD ratios of $\sim 0.6$ over the well known shipping lane leading from South India to the Strait of Malacca. The shipping signal in this region was first identified in satellite $NO_2$ data by (Beirle et al., 2004; Richter

et al., 2004). Here, while the shipping lanes are clearly visible in the vis SCDs, they cannot be identified in the UV $NO_2$ data. This is probably because the $NO_2$ shipping emissions are located close to the ground. Consequently, the shipping emissions when averaged over the ground scene are below the detection limit for the UV spectral range (see Fig. 6). Higher horizontal resolution (e.g., in OMI and the upcoming S-5P, S4, and S-5 missions) might improve detection of shipping lanes in the UV.

## 3.2   Temporal variability of regional $NO_2$ slant columns

In this section, we investigate the temporal variability of tropospheric SCDs from the two $NO_2$ retrievals over six regions (see Tab. 2 and Fig. 7 for the regions' definitions) and make inferences about the vertical distribution of $NO_2$ in the atmosphere. Figure 9 shows monthly mean time series for three natural / biomass burning source regions (a – c) and three anthropogenic source regions (d – f). The seasonal cycle observed in Fig. 9 a – c corresponds to the

seasonal pattern of biomass burning activity in these regions (Schreier et al., 2014). Slightly negative SCDs in the UV fit in north Australia (NAUS; Fig. 9 c) and Riyadh (Kingdom of Saudi Arabia; Fig. 9 f) are artefacts caused by the stratospheric correction (see Sect. 2.5). Another artefact in the data is related to degradation of the GOME-2A instrument (see Sect. 2.7; Dikty and Richter, 2011), which is in some regions represented by a slightly decreasing linear trend (for example: in the UV in ANE and HPSA; Fig. 9 b and c).

[Table 2 about here.]

                                  [Figure 9 about here.]

In both ASE and NAUS, the seasonal cycles in the UV and vis spectral range are similar (Fig. 9 a and c), but the tropospheric $NO_2$ SCDs in the UV are smaller than in the vis spectral range. As can be seen in Fig. 10, this results in




the strong correlation coefficients (0.87 in both cases, see Tab. 3) and slopes $\leq 1$ of the regression lines (see Tab. 4). When the seasons are considered individually, the slope differs between 0.65 (biomass burning season) and 1.04 (rainy season). Hence, for both regions in the rainy season the two spectral ranges differ only by an offset whereas in the biomass burning season large differences are observed. Furthermore, in both regions for the individual seasons the correlation coefficient is also high $(0.50-0.99$; see Tab. 3), especially in the biomass burning season $(\geq 0.92)$, which is probably related to the higher signal to noise ratio for this season. In ANE, conversely, significant differences in the seasonal cycle between the two spectral ranges can be observed, as the $NO_2$ signal from the biomass burning peak in NH winter shows in the vis spectral range (Fig. 9 b) only while in the UV, no interannual variability can be detected. The increased $NO_2$ load visible in the vis SCDs cannot be detected in the UV SCDs (correlation coefficient is only 0.75). In all the other seasons, the correlation coefficient is comparable to that observed in ASE and NAUS (see Fig. 10 b and Tab. 3).

[Figure 10 about here.]

[Table 3 about here.]

Several effects could contribute to an explanation for the differences in visibility of $NO_2$ in ANE in the two spectral ranges:

1. Cloud influence on $NO_2$ measurements: When considering only those measurements flagged as cloudy (i.e., having a cloud coverage $\geq 0.3$), only a weak seasonal cycle can be found in ANE (Fig. A1, App. A1), whereas ASE shows a similar seasonal cycle in the cloud covered case as in the cloud free case. This indicates that the biomass burning $NO_2$ in ASE might be located partially in elevated $NO_2$ layers above the clouds, whereas in ANE $NO_2$ is located closer to the ground, i.e., usually below and thus partly shielded by the clouds.

2. $NO_2$ layer altitude: The altitude of biomass burning $NO_2$ emissions influence their visibility (see Fig. 2). According to simulations with the TM5-model (Williams et al., 2017; for details, see App. A2), the layer height of biomass burning $NO_2$ over ANE is lower than over ASE (Fig. A2). This is consistent with the above result, which shows high and low visibility for the biomass burning $NO_2$ in the UV for the ASE and ANE regions, respectively. For ASE, several previous studies could show that biomass burning plumes are regularly located at least partly above the boundary layer (Coheur et al., 2007; Rio et al., 2010; Gonzi and Palmer, 2010).

3. Stratospheric correction: For observations in the UV spectral range, total SCDs (i.e., the SCDs resulting from the DOAS fit, without the stratospheric correction described in Sect. 2.5) over ANE show two distinct peaks over the course of the year: one peak in winter and another peak in summer (Fig. A3, App. A3). The winter peak in Dec / Jan falls in the main biomass burning season and is thus expected to be of tropospheric origin. In contrast, the summer peak in May – Jul is of stratospheric origin, as stratospheric $NO_2$ has its maximum




in summer. It is noteworthy that in the vis spectral range, this summer peak is smaller compared to the UV, which might be partly related to differences in the stratospheric sensitivity between the two wavelength ranges. In the UV SCDs, the biomass burning $NO_2$ might thus be obscured by the similar magnitude of the stratospheric summer peak.

4. Cloud influence on data sampling: The data shown in Fig. 9 are filtered to include only measurements not strongly influenced by clouds. In satellite-based cloud retrievals, smoke is often misinterpreted as cloud (Boersma et al., 2004). Assuming that smoke and $NO_2$ from biomass burning is advected together, this could lead to large parts of the biomass burning $NO_2$ being filtered out due to apparent cloud contamination of the measurement. However, the vis $NO_2$ SCDs are clearly enhanced in the cloud-free measurements, showing that not all biomass burning $NO_2$ is filtered out. Furthermore, there are only small differences in the cloud coverage between ANE and ASE (not shown); in both regions, the number of pixels being filtered out due to cloud contamination is similar. The minimum fraction of cloud free pixels is about $30\%$, observed during the rainy season, while the maximum fraction of about $70-80\%$ and $80\%$ in ANE and ASE, respectively, is observed during the dry season, when the biomass burning occurs. It seems thus unlikely that the cloud filtering significantly impacts the visibility of the biomass burning $NO_2$ over ANE.

5. Aerosols: The $NO_2$ measurements could be influenced by aerosols. According to CALIPSO measurements, the main aerosol-type in ASE is smoke from biomass burning, while other aerosol types contribute only a smaller amount to the total aerosol concentration (Fig. A4, App. A4). In contrast, the dominant aerosol-types in ANE are dust and polluted dust. As in the UV spectral range, the single scattering albedo (SSA) of dust is smaller than that of biomass burning aerosols (Russell et al., 2010; Dubovik et al., 2001; Bergstrom et al., 2007), the biomass burning $NO_2$ signal in ANE could be shielded by the darker dust aerosol in the UV. However, this hypothesis seems unlikely, as the number of dust aerosols (per area) is even higher in Riyadh compared to ANE, while in Riyadh a clear $NO_2$ seasonal cycle can be identified in both spectral ranges.

In summary, we believe that the absence of biomass burning $NO_2$ signal in the UV spectral range over ANE is predominantly formed by the lower injection height, causing the $NO_2$ to be located lower in the atmosphere compared to ASE. Due to the strong differences in sensitivity between UV and vis spectral ranges close to the ground, this could to a large extent explain the invisibility of UV $NO_2$ in ANE. However, also further effects could influence the visibility. For example, interferences of the seasonal cycles of stratospheric and tropospheric $NO_2$ over ANE might contribute, while we deem the effect caused by differing prevalent aerosol types in the three regions unlikely to be significant.

Figures $9\,d-f$ show three regions dominated by anthropogenic air pollution. Here, absolute differences between UV and vis $NO_2$ values are often larger than in biomass burning source regions (see also Fig. 8). The seasonal cycle is mostly larger in the vis spectral range compared to the UV spectral range, indicating a seasonal dependency of





the reasons underlying the differences between the spectral regions. In general, the time series of the two spectral ranges show a similar behaviour and are highly correlated (0.82; Tab. 3).

   Over East China, the $NO_2$ SCDs in UV and vis have similar shape and show a very high correlation of $\geq 0.98$ (Fig. 9 d). Until 2012, increasing $NO_2$ winter values in both spectral ranges can be observed; these increases have

been analysed in depth by Richter et al. (2005) and Hilboll et al. (2013a). In 2013, there are nearly no changes in $NO_2$ and afterwards, $NO_2$ SCDs are slightly decreasing, which was already reported by Richter et al. (2015) and Irie et al. (2016) and is consistent with findings reported by Hilboll et al. (2017) for anthropogenic $NO_2$ pollution in India. Over the HPSA region, no $NO_2$ trend is observed (Fig. 9 e). A clear seasonal cycle is observed in both spectral ranges, with the highest correlation coefficient in SH winter (0.90; Tab. 3). The detected seasonal cycle in the vis

spectral range is similar to the cycle found by Noije et al. (2006) in the year 2000 in data from the Global Ozone Monitoring Experiment (GOME). In Riyadh (Fig. 9 f) a small trend is observed in maximum values. As shown by Hilboll et al. (2013a), during the whole period a slight increase can be detected. The detected month with maximum $NO_2$ is in agreement with van der A et al. (2008). In this area the $NO_2$ values derived from the UV and vis spectral range show a similar seasonal cycle and a similar year-to-year variability with a correlation coefficient of $\geq 0.95$.

[Table 4 about here.]

   In anthropogenically polluted areas, the differences in the $NO_2$ signal strength between the two spectral ranges are a result of the larger sensitivity to the lower troposphere in the vis spectral range. In winter the air pollution is mostly stronger, compared to summer, the SZA is lower, the boundary layer is more shallow (von Engel and Teixeira, 2013), and even within this boundary layer also a strong gradient is expected over high-emission areas

(Dieudonné et al., 2013). All these effects increase the observed differences between the two spectral ranges. Due to the higher profile sensitivity close to the ground and the resulting larger differences, a lower regression slope (see Fig. 10) is observed for the anthropogenically polluted areas than for the biomass burning regions (Tab. 4) in all seasons. For example, in NAUS and ASE the regression line between all UV and vis SCDs has a slope of 0.66 and 0.74, respectively, whereas for China and HPSA, a considerably smaller slope ($\sim 0.4$) is found.

One has to note that the regions ANE and Riyadh do not follow this pattern — ANE is attributed the missing of the biomass burning $NO_2$ signal in the UV, and Riyadh probably results from differences in boundary layer height and the maximum in $NO_2$ seasonal cycle (von Engel and Teixeira, 2013). In Riyadh, the seasonal cycle of $NO_2$ emission and the cycle of the boundary layer height are in phase as it is the case for naturally polluted areas, whereas for China and HPSA, the $NO_2$ maximum can be found in the respective winter season, where a lower

boundary layer height is observed. Consequently, in Riyadh the UV $NO_2$ signal is stronger and the slope (0.74) is larger than usually expected for anthropogenic source regions.

   Finally, a seasonal dependency of the regression line slope may result from the combination of the boundary layer height and the surface concentration of $NO_2$. For ASE and NAUS, the slope shows a seasonal pattern, which might be influenced by the small signal to noise ratio (see Tab. 4). Slopes of $\sim 1.0$ can be found in these areas for the rainy



season, with the largest uncertainties for these areas, possibly related to high cloud cover. The differences are thus most likely introduced by an offset. During biomass burning season, the slope is about 0.83 and 0.71 for ASE and NAUS, respectively. Since the biomass burning $NO_2$ signal cannot be seen in UV SCDs over ANE, no clear seasonal pattern is visible there. In East China and the HPSA region, a larger slope can be found in NH summer, when the

5 boundary layer is higher; spring, autumn and winter show a smaller slope. This is expected due to the increased sensitivity (see Fig. 2) of the UV measurements to $NO_2$ in a thick compared to a shallow boundary layer. In Riyadh however, this effect is not observed, possibly due to a decreased signal to noise ratio caused by the smaller area, entailing fewer measurements or a stronger stratification of the boudary layer.

### 3.3 Spatial distribution of $NO_2$ vertical columns

The $NO_2$ vertical column densities (VCDs) are the final result of the DOAS retrieval procedure, as they are a physically meaningful and universally comparable quantity. In this section, we compare $NO_2$ tropospheric VCDs retrieved from the UV and the vis spectral range. For the calculation of the VCDs, $NO_2$ vertical profiles from the TM5-model (Williams et al., 2017) are used as a-priori (see Sect. 2.6). Assuming perfect measurements, radiative transfer simulations, and a-priori profiles, the AMF would equalise all differences between the two spectral ranges,

leading to identical VCDs for UV and vis fits. Figure 11 shows maps of monthly means $NO_2$ VCDs for January (upper panels) and July (lower panels) 2008, retrieved from the UV (a, c) and vis fits (b, d). The spatial patterns in both datasets are very similar and agree well with those visible in SCD maps (Fig. 6).

[Figure 11 about here.]

However, also the VCDs of both spectral ranges do not perfectly match, as illustrated in Fig. 12. Compared to

20 the SCD differences shown in Fig. 7, the VCD differences are smaller. Especially over areas without significant $NO_2$ sources, but also over the biomass burning regions, the differences between the two retrievals are reduced compared to the SCDs. Over anthropogenically polluted areas however, e.g., over China in January, the VCDs retrieved from the two spectral regions still show significant differences. The most probable reason for this is that the simulated $NO_2$ profiles used as a-priori do not represent the actual $NO_2$ vertical distribution; furthermore, aerosols and the

25 SSR might influence the calculated VCDs, as will be discussed in Sect. 3.4.

[Figure 12 about here.]

### 3.4 Temporal variability of regional $NO_2$ vertical columns

To better understand the reasons for the differences between the tropospheric VCDs of the two spectral regions, we again investigate the temporal variability over the six regions shown before (see Sect. 3.2, Tab. 2, Fig. 7 for the

30 region definitions). The general shape of the seasonal cycles, which was described for SCDs in Sect. 3.2, can also be found in the VCD time series (Fig. 13). Similar to the SCD time series (see Sect. 3.2), the $NO_2$ VCDs are still





mostly larger in the vis spectral range. Compared to the SCD differences, however, the VCD differences between the two spectral ranges are reduced in all six regions (Fig. 13) which is also visible in the higher slope of the regression line (Tab. 5). For nearly all regions and seasons, the slope of the regression line (Tab. 6) is still below one, indicating that the calculated AMFs are not representative for the actual state of the atmosphere. The SSR and the relative
vertical profile as well as aerosols may contribute to the differences of VCDs between the two spectral ranges.

In the ASE region, for example, the seasonal cycle in the VCDs is smoother than in the SCDs, especially for the UV data (Fig. 13 a). The seasonal cycles of the VCDs in the two spectral regions are more parallel, leading to a high correlation coefficient of 0.96 between the two data sets (see Tab. 5). However, rather than being separated by a simple offset, the VCDs from the two spectral ranges seem to be linearly related as indicated by the regression
slopes differing from one (see Tab. 6 and Fig. 14). Compared to SCDs, the slope for the VCDs increases slightly in nearly every season; for the annual values, the slope of VCDs is 0.70 compared to 0.66 for SCDs.

[Figure 13 about here.]

[Table 5 about here.]

In the ANE region, the VCDs from the UV and vis spectral ranges agree considerably better than for the respective
SCDs (Fig. 13 b). The UV $NO_2$ VCDs show a clear seasonal cycle, of similar shape but reduced amplitude compared to the vis VCDs, leading to reduced differences between the two spectral ranges. Comparing to the SCDs in the ANE region, this indicates that the biomass burning $NO_2$ seems to be located too close to the ground to be detected by the UV retrieval; however, as the $NO_2$ emitted by fires is included in the simulation from which the a-priori profiles are taken, the AMFs introduce the observed seasonal cycle in the UV VCDs. This is reflected by the considerably
improved correlation coefficients of 0.91 for all VCDs (see Tab. 5) compared to 0.53 for all SCDs; similar improvements can be seen for all seasons individually. Interestingly, when the seasons are considered separately, the regression line has higher slopes between 0.72 (SON) and 0.98 (JJA), showing that in the rainy season, the UV and vis VCDs only differ by an offset. Nevertheless, discrepancies between the two spectral ranges are still observed, especially during biomass burning season.

For NAUS, the correlation coefficient between UV and vis VCDs is higher (0.94) than for the SCDs (0.78). Also in this area, the slope of the VCD regression lines is constant or increased compared to the SCDs (Tab. 6; Fig. 14). During rainy season, the observed differences between UV and vis VCDs are mostly related to an offset, whereas during biomass burning season the slope is smaller than one.

For VCD time series of East China (Fig. 13 d), the upward trend which was visible in the winter $NO_2$ SCDs (see
Fig. 9 d) is not present in the VCDs. However, also here the UV $NO_2$ values are still smaller than the vis $NO_2$ values. In spite of these discrepancies, the correlation coefficient is highest for this region ($\geq 0.96$ in all seasons). During all seasons the correlation coefficients are increased or similar compared to those of the SCDs. While the slope of the regression line is still smaller than one ($0.54 - 0.69$; Tab. 6), it is still increased compared to SCDs.





The Highveld (HSPA, Fig. 13 e) region differs from all other regions: a high correlation coefficient (0.90) can be found in SH winter, when also the slope of the regression line (0.92) as well as the offset is high. In SH summer, however, the smallest correlation coefficient is observed for the region (0.78) and also the slope is the smallest with 0.69.

In Riyadh, large relative differences between the two spectral ranges can be found compared to the other regions (Fig. 13 f). This is reflected by the slope of the regression line which is $\sim 0.65$, except for NH winter when the slope is 1.09 with a large offset (Fig. 14 f).

[Figure 14 about here.]

[Table 6 about here.]

The observed discrepancies between the two spectral ranges can result from different reasons: they can be related to the a-priori profiles used for VCD calculations, or they can be related to other influencing factors like aerosols and surface properties. The importance of the a-priori profile shape for the resulting VCDs was shown by Martin et al. (2002) and Boersma et al. (2004); the resulting uncertainty for this parameter is assumed to be smaller than 15 % (Boersma et al., 2004). A potential error in the a-priori profiles from the TM5-model in a given altitude layer

will lead to different errors in the VCDs calculated for the two spectral ranges, due to the spectral dependence of the measurement sensitivity (i.e., the a-priori error is weighted with different sensitivities in the two spectral ranges). This is especially relevant in biomass burning season, when the quality of the a-priori profiles is probably lower than usual, since the modeling of biomass burning emissions entails large uncertainties due to spatial resolution, emission uncertainties, and assumptions made on the plume injection height (see, e.g., Archer-Nicholls et al., 2015 and

references therein). Similarly, since our retrieval does not explicitly account for the effect of aerosols, an aerosol layer at a given altitude will influence the visibility of the $NO_2$ both above and below the aerosol differently, enhancing the difference in the VCDs retrieved from the two spectral ranges. The same is true for potential errors in the assumed SSR. As shown by Boersma et al. (2004), in March the uncertainties of $NO_2$ concentration are largest in polluted areas with low SSR, but they are generally below 50 %.

Tropospheric model VCDs are plotted in Fig. 13, showing a regional dependency of the discrepancies between the model VCDs and the observed VCDs. For ASE and ANE during biomass burning season, the UV $NO_2$ VCDs and the model VCDs agree quite well, while the vis $NO_2$ values are higher. In rainy season, especially in ANE, there are larger discrepancies between the UV and vis $NO_2$ and the model VCDs. The model values are lower though the UV and vis spectral ranges agree quite well. This confirms the above finding that the differences between the

spectral ranges are caused by an offset. The differences during biomass burning season might be related to the aerosol treatment or wrong SSR in AMF calculations.

Also for NAUS the simulated and observed values agree quite well in the rainy season, whereas in the biomass burning season large differences between the three values are observed. Therefore, in this regions a combination of the different factors, described above, will most likely provide the best explanation.



For China and Riyadh, in the less polluted season the model and UV $NO_2$ values agree quite well whereas the vis $NO_2$ values are larger. In the season with high air pollution, in both regions large discrepancies between the three values are visible. For anthropogenically polluted areas, the discrepancies between the modelled and the measured $NO_2$ values are well known from previous studies (e.g., Huijnen et al., 2010). For these regions, also aerosols play an important role and can further influence the differences between the two spectral ranges. As shown by Russell et al. (2010), especially dust reduces the visibility in the UV spectral range which could partly explain the differences in Riyadh. Additionally, as shown by Boersma et al. (2004), for these anthropogenically polluted areas also the SSR can introduce large differences.

In HPSA, an offset between the model and the observed values is clearly visible, which could partly explain the differences in SH winter values (Tab. 6). However, also here the SSRs and aerosols might influence the measurements, and therefore, the calculation of AMF.

## 4  Summary and Conclusion

In this study, we present a new $NO_2$ DOAS retrieval for measurements by the GOME-2A instrument, using the ultraviolet (UV) spectral range. Having $NO_2$ measurements from this wavelength range in addition to the usually exploited visible (vis) spectral region gives rise to conclusions about the vertical distribution of tropospheric $NO_2$, as the vertical sensitivity of space-borne $NO_2$ DOAS retrievals strongly depends on the wavelength.

$NO_2$ slant column densities (SCDs) can in principle be retrieved from the UV spectral range, as shown by Yang et al. (2014). However, the UV retrievals have larger uncertainty compared to the vis spectral range, caused by the smaller differential absorption and the reduced intensity in the UV spectral range, leading to inferior fit quality and, also due to the larger AMF uncertainties. Nonetheless, the spatial distribution of both retrievals agrees very well.

We focused our analysis on six regions: three biomass burning regions (Africa south of the equator / ASE, Africa north of the equator / ANE, and northern Australia / NAUS), and as three anthropogenic source regions (Eastern China, the Highveld Plateau in South Africa / HPSA, and the surroundings of Riyadh).

The differences between the UV and vis fits are lower over the biomass burning areas ASE, ANE, and NAUS compared to the regions dominated by anthropogenic pollution. The $NO_2$ from biomass burning is in some regions and periods located in elevated layers, leading to an effectively smaller sensitivity difference between UV and vis spectral ranges. This is reflected by the slope of the regression lines between UV and vis SCDs, which is 0.66 and 0.74 for the ASE and NAUS regions, respectively. Anthropogenic air pollution on the other hand is mostly located in the boundary layer close to the ground (see Stull, 1988). As the sensitivity of the UV retrieval drops sharply towards the surface, this leads to considerably lower regression line slopes of 0.37 and 0.41 for East China and HPSA, respectively. Nonetheless, SCDs in the UV and vis show high correlation coefficients $\geq 0.82$ in both cases.

In ANE, the biomass burning $NO_2$ in NH winter is not visible in the UV SCDs. Our analysis of UV $NO_2$ SCDs over cloudy scenes and of $NO_2$ profiles simulated by the TM5-model both suggest that in ANE the $NO_2$ located close



to the ground, where sensitivity in the UV is limited. The invisibility could be increased by enhanced stratospheric $NO_2$ over the Pacific which counteracts with the biomass burning $NO_2$ over ANE and therefore balance enhanced SCDs. Moreover, the ANE biomass burning $NO_2$ could be shielded by the predominantly dust aerosols, which are dark in the UV and thus lead to decreased sensitivity compared to other biomass burning regions. In ASE, for

example, the predominant aerosol type is smoke, which is brighter than dust in the UV spectral range; combined with the higher altitude of the $NO_2$ this could lead to increased visibility in ASE.

In Riyadh, the seasonal cycles of boundary layer height and tropospheric $NO_2$ are in phase, contrary to other anthropogenically polluted areas, where $NO_2$ columns are highest in winter, when the boundary layer is low. There-fore, the maximum $NO_2$ values coincide with high boundary layers and thus better visibility in the UV fit, leading

to better agreement between SCDs from the two spectral regions, expressed by a higher slope of the regression line of 0.74.

Even though in theory, given perfect measurements, radiative transfer modelling, and a-priori information, ver-tical column densities (VCDs) retrieved by the DOAS technique should be independent on the wavelength, VCDs retrieved from the UV and vis fits do still show considerable differences. If the a-priori profile does not capture

the actual vertical distribution of $NO_2$ in the troposphere, the vertical measurement sensitivity will be incorrectly aggregated in the air mass factor calculation which will lead to wavelength-dependent VCDs. One, admittedly un-likely, explanation would be if the model over-estimated vertical mixing, leading to an underestimation of the AMF UV / vis ratio. Having said this, the observed VCD differences can also be introduced by further input parameters like surface spectral reflectance (SSR; wavelength: 354 nm and 442 nm) and the lack of explicit aerosol treatment. These

parameters can lead to large discrepancies between the two spectral ranges when assumed wrongly. Even though differences exist between the two spectral ranges, the correlation coefficient between the VCDs is high everywhere ($\geq 91\%$). Generally, the differences between the two spectral ranges indicate that the $NO_2$ in the atmosphere is located in lower atmospheric layers than in the model which has stronger inferences in the UV or SSR is to high in the vis spectral range.

While vis $NO_2$ VCDs are often larger than UV VCDs, the latter agree with the simulated VCDs in most seasons. For China and HPSA, the differences between the two spectral ranges are pronounced in the respective winter season, whereas for Riyadh large differences can be found throughout the whole year. However in HPSA, in SH winter the correlation coefficient and the slope of the regression line is largest with an high offset. In the biomass burning regions, the agreement is better for the rainy seasons; during dry season, when a larger aerosol load is expected, the

differences are larger. After conversion into VCDs, in ANE a clear seasonal cycle can be observed in both spectral ranges, which is mostly introduced by the a-priori assumptions in the AMF calculation.

It is currently not possible to retrieve vertical profiles of tropospheric $NO_2$ from satellite measurements, due to the low information content of the measured spectra (see, e.g., Hilboll et al., 2014). Further developments in space-based observation of the Earth's atmosphere could however increase the capability to retrieve vertical $NO_2$ distribution.

For example, concurrent measurements of the same air mass from different viewing geometries could yield insight



on aerosol types and vertical profiles. This in turn would increase the possibility to retrieve more quantitative information about the vertical distribution of $NO_2$ in the troposphere.

However, our study shows that even for the current generation of instrumentation in the global observing system, the difference between vis and UV $NO_2$ DOAS retrievals can be used to constrain our knowledge of $NO_2$ vertical dis-

tribution in the lower troposphere. For example when coupled with data assimilation techniques and high resolution models the amount of boundary layer $NO_2$ and its trends can be inferred more accurately.

## Appendix A: Shielding of northern hemispheric winter values in Africa north of the equator

### A1    Shielding by clouds

Figure A1 shows $NO_2$ time series for cloud free scenes ($\leq 20\%$ cloud fraction; a and c) and for cloudy scenes ($\geq 30\%$

cloud fraction; c and d). The decrease / increase of $NO_2$ values in recent years visible in Fig. A1 is probably related to instrumental degradation (Dikty and Richter, 2011); a similar decrease / increase cannot be observed in GOME-2B data (not shown). In ASE, the seasonal cycles for cloud covered and cloud free measurements are similar. In ANE, however, only a weak seasonal cycle can be found for the cloudy scenes in either spectral range, whereas in the cloud free scenes a clear seasonal cycle can be found in the vis but not in the UV spectral range. This implies that the

$NO_2$ layer over ASE could be located also in higher altitudes, while over ANE the $NO_2$ could be located closer to the ground and thus less visible to the UV observations.

[Figure A1 about here.]

### A2    $NO_2$ profile shape

To compare the vertical distribution of $NO_2$ over the biomass burning areas ANE, ASE, and NAUS, simulated

$NO_2$ concentration fields from the TM5-model dataset (Williams et al., 2017) for 2008 at 09:00 LT were used. We assume that the small time difference between the satellite measurements and the model output can be neglected. Figure A2 (a) shows profiles of $NO_2$ concentration for the regions ANE and ASE for four months. Apparently, $NO_2$ is located at higher altitudes in ASE, which leads to differences in the visibility of the $NO_2$ signal between the two spectral ranges. This is partly related to the surface altitude which is on average $\sim 830\,$m above sea level in ASE

and only $\sim 440\,$m in ANE. This could explain why the UV fit shows $NO_2$ over ASE and not over ANE.

[Figure A2 about here.]

### A3    Seasonality of stratospheric $NO_2$

Figure A3 shows time series of detrended total SCDs for the regions ANE and ASE. Detrending is necessary because of the degradation of the instrument (Dikty and Richter, 2011), and is implemented as subtraction of a linear trend.





Here, a shortened time series from 2008 to 2012 is used, since changes in the instrument configuration (EUMETSAT, 2015) might introduce an offset after 2012.

In ASE a clear seasonal cycle can be seen for both spectral ranges (Fig. A3 a), which agrees well with the seasonal cycle expected from biomass burning activity. However, comparison with Fig. A3 c shows that a similar seasonal
cycle can also be observed in SCD data over the Pacific Ocean (averaged over $180° - 210°$ E), which is void of any $NO_x$ emissions and should therefore only have very low tropospheric $NO_2$ SCDs. This indicates that the seasonality over ASE is at least partly caused by stratospheric fluctuations.

Over ANE, slightly different seasonal cycles can be observed in the UV and vis spectral range (see Fig. A3 b). The peak related to biomass burning in NH winter is clearly observed in the vis SCD data. In some years a small
secondary peak in NH summer can be noticed. Similar seasonal variability can be seen in the UV spectral range, but both peaks have similar magnitude there. This double peak structure can also be found in $NO_2$ SCDs of the Ozone Monitoring Instrument (OMI, not shown). Over the Pacific Ocean (Fig. A3 d), the second peak in NH summer is clearly visible in the UV data, corresponding to the known seasonal variability of stratospheric $NO_2$. As the seasonal cycles of biomass burning and stratospheric $NO_2$ are in-phase and out-of-phase for ASE and ANE, respectively, the
visibility of the biomass burning $NO_2$ differs strongly between the two regions.

[Figure A3 about here.]

## A4   Aerosol effect

The Cloud-Aerosol Lidar and Infrared Pathfinder Satellite Observations (CALIPSO) satellite is in a sun-synchronous polar orbit with an equator crossing time of 13:30 LT (Winker et al., 2007). It was launched in April 2006 and has a
repeat cycle of 16 days. On CALIPSO, the Cloud-Aerosol Lidar with Orthogonal Polarisation (CALIOP) instrument is operating since June 2006. CALIOP is a nadir-viewing two wavelength polarization-sensitive lidar, operating at 532 nm and 1064 nm. The horizontal and vertical resolution depends on the altitude. Closer to the ground the resolution increases in both cases (Winker et al., 2007; Winker et al., 2009).

Here, the CALIOP level 3 product "Aerosol Profile All Sky (daytime)" was used. The level 3 data are on a 2°
latitude times 5° longitude grid (Atmosheric Science Data Center, 2009). For the calculation of regional aerosol contribution for smaller regions, the amount of aerosols in the grid box was weighted by the contribution of the grid box to the region. This dataset provides the six aerosol types "dust", "polluted continental", "smoke", "clean marine", "clean continental", and "polluted dust", where the latter accounts for a mixture of dust and smoke or a mixture of dust and urban pollution (Omar et al., 2009). The aerosol-types were derived using aerosol models based
on cluster analysis of an AERONET dataset and the aerosol extinction-to-backscatter ratio (Omar et al., 2009). In the present study, the differences in equator crossing time can be ignored, because especially for ASE and ANE larger areas are averaged.

Figure A4 shows the sample number for the six aerosol types over the three regions ANE, ASE, and Riyadh. The dominant aerosol-type in ASE is smoke whereas dust dominates in ANE and Riyadh.



[Figure A4 about here.]

*Acknowledgements.* This study has been funded by the EU FP7 project PArtnership with ChiNa on space DAta (PANDA, grant no. 606719), by DLR in the scope of the Sentinel-5 Precursor verification projekt (grant no. 50EE1247), by the University of Bremen, the state of Bremen. GOME-2 lv1b radiances have been provided by EUMETSAT. The CALIOP data were

5   obtained from the NASA Langley Research Center Atmospheric Science Data Center.



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

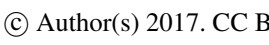



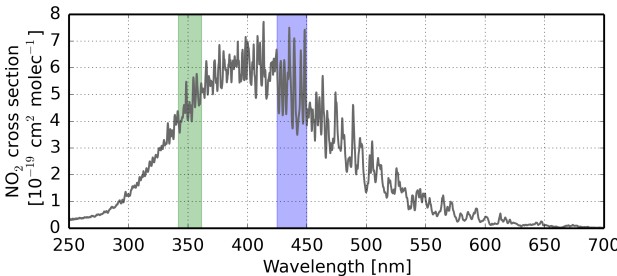

**Figure 1.** NO$_2$ absorption cross section measured at 243 K with the GOME-2 instrument now deployed on MetOp-A. The fitting windows used in this study are shaded by color. Green: new NO$_2$ fitting window in the UV. Blue: NO$_2$ fitting window in the vis spectral range (see Tab. 1).

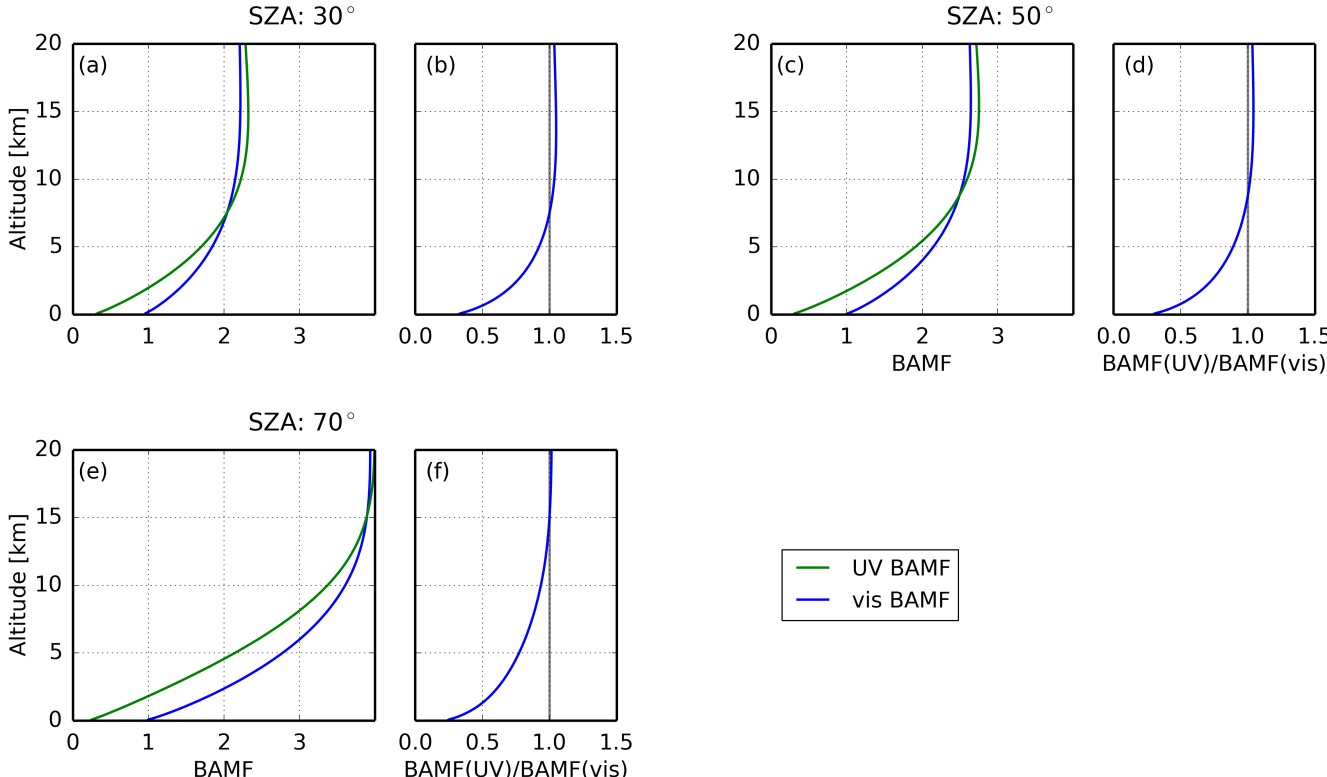

**Figure 2.** BAMF for UV (green line) and vis (blue line) spectral range and the ratio of both BAMFs calculated with the radiative transfer model SCIATRAN. BAMFs converge with higher altitude (not shown). The BAMFs are calculated for 352 nm (UV) and for 438 nm (vis) at (a, b) 30° (c, d) 50° and at (e, f) 70° solar zenith angle (SZA). A surface spectral reflectance of 0.04 (UV) and 0.06 (vis) is assumed.




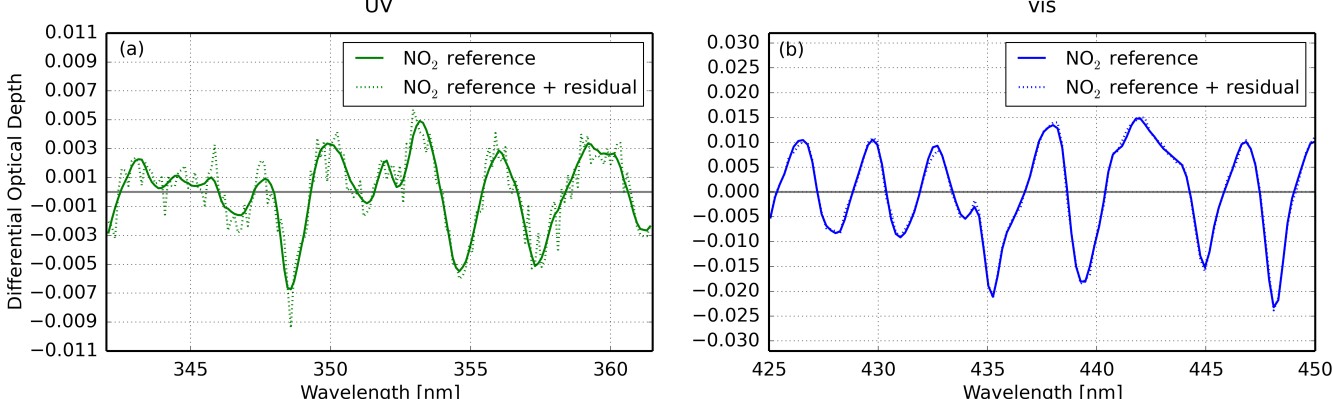

**Figure 3.** The $NO_2$ reference ($NO_2$ cross section multiplied with retrieved $NO_2$ SCD; solid line) and reference plus residual (dashed line) for the (a) UV and (b) vis spectral range for one pixel above Teheran ($35.38°$ N, $51.47°$ E) on 22 January 2008. The SCD for this pixel for the UV spectral range is $6.31 \times 10^{16}$ molec cm$^{-2}$ with a fit error of $4.3\%$. The SCD for this pixel for the vis spectral range is $9.33 \times 10^{16}$ molec cm$^{-2}$ with a fit error of $0.8\%$. Note the different y-axes.

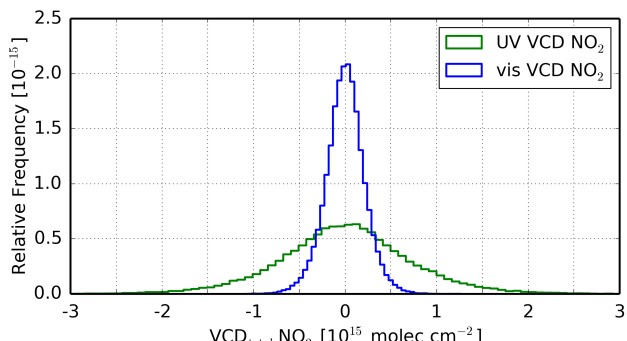

**Figure 4.** Distribution of total $NO_2$ VCDs over a clean region (equatorial Pacific: $5°$ S $- 5°$ N and $150°$ E $- 210°$ E) for the UV and vis spectral range for January 2008. Curves are normalised to unit area and centered on zero.





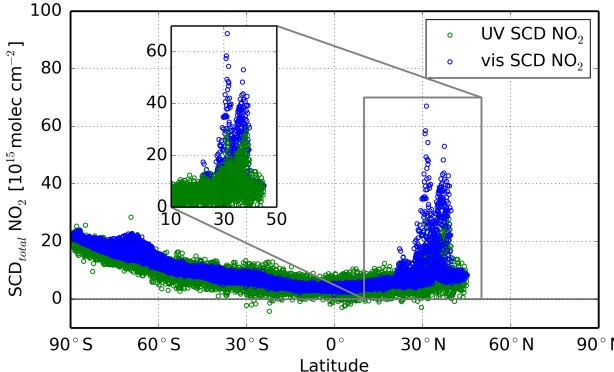

**Figure 5.** Total NO$_2$ SCDs of one orbit (above East China on 03 January 2008) for the UV und vis spectral range. Only data with SZA smaller 70° are shown.

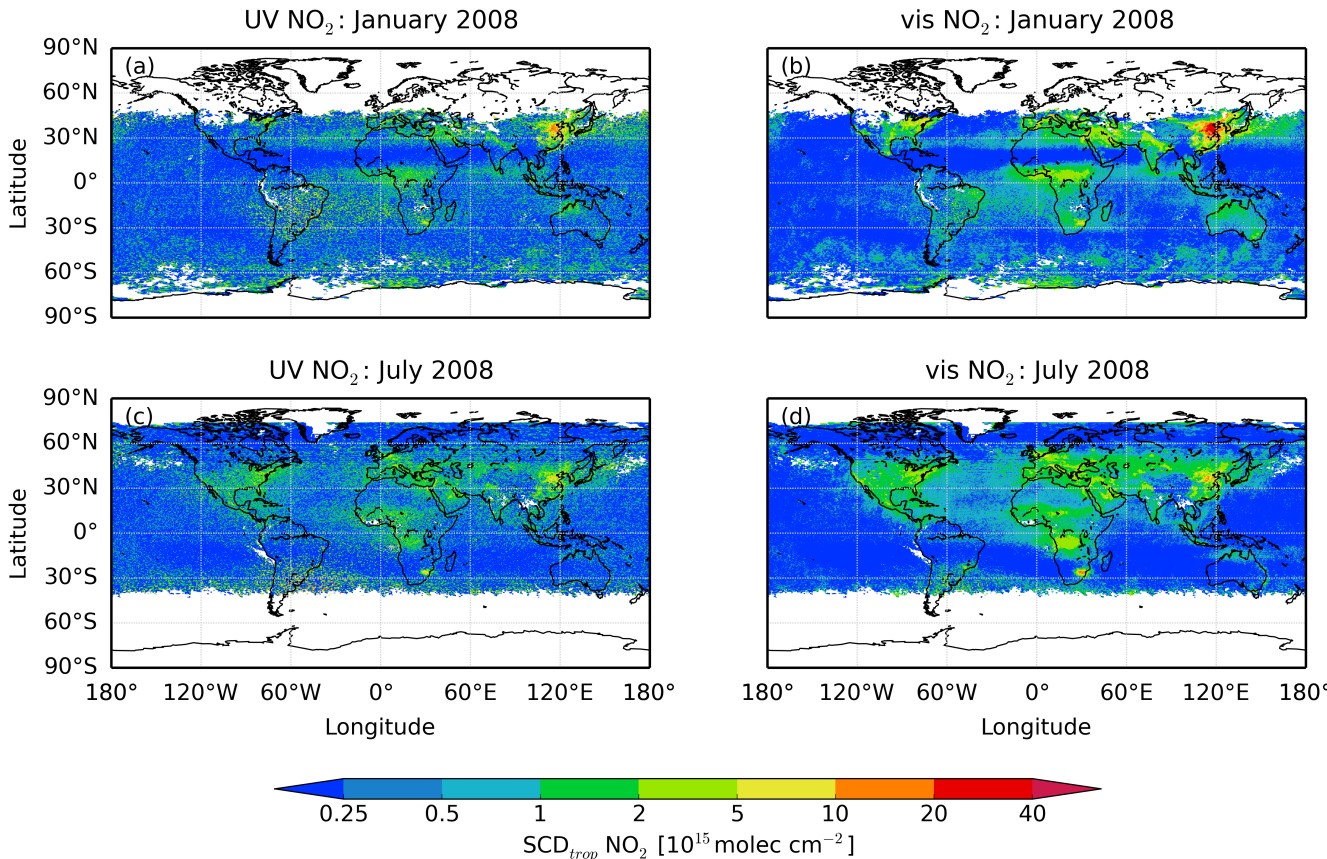

**Figure 6.** Monthly mean tropospheric NO$_2$ SCDs for (a, b) January and (c, d) July 2008. (a, c) UV spectral range and (b, d) vis spectral range.





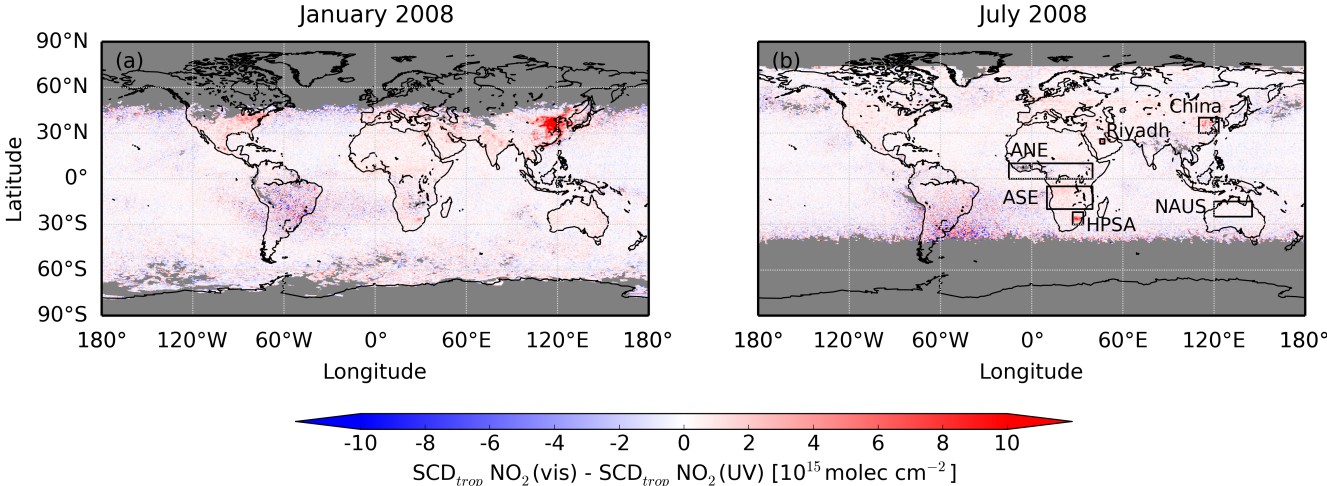

**Figure 7.** Absolute difference between monthly mean tropospheric $NO_2$ SCDs in the vis and UV spectral range. Differences for (a) January and (b) July 2008. Dark gray shaded area: no $NO_2$ values available. The regions defined in Tab. 2 are depicted in the right figure (b).

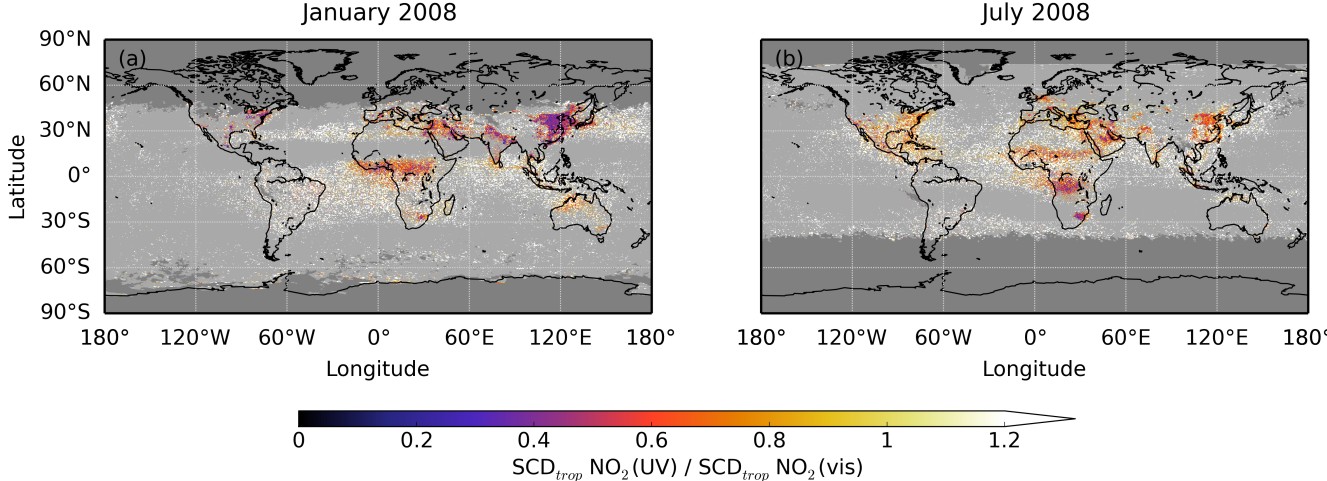

**Figure 8.** Ratio between monthly mean tropospheric SCDs of $NO_2$ in the UV and vis spectral range. (a) January 2008 and (b) July 2008. Dark gray shaded area: no $NO_2$ values available. Light grey coloured values indicate values where the vis $NO_2$ is close to zero, which have been filtered out.





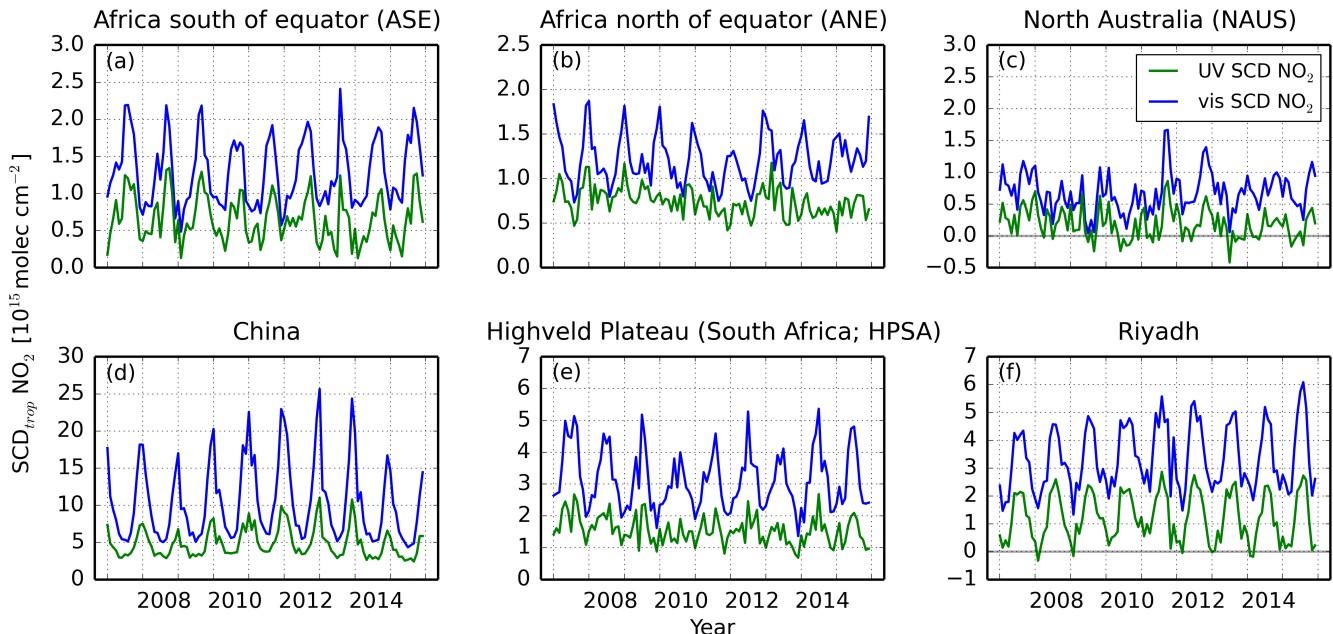

**Figure 9.** Time series $2007-2015$ of tropospheric $NO_2$ SCDs for the UV (green line) and vis (blue line) spectral range for different regions. Note the different y-axes. The vertical lines indicate January of the individual years. (a$-$c) biomass burning regions and (d$-$f) regions with high anthropogenic air pollution.





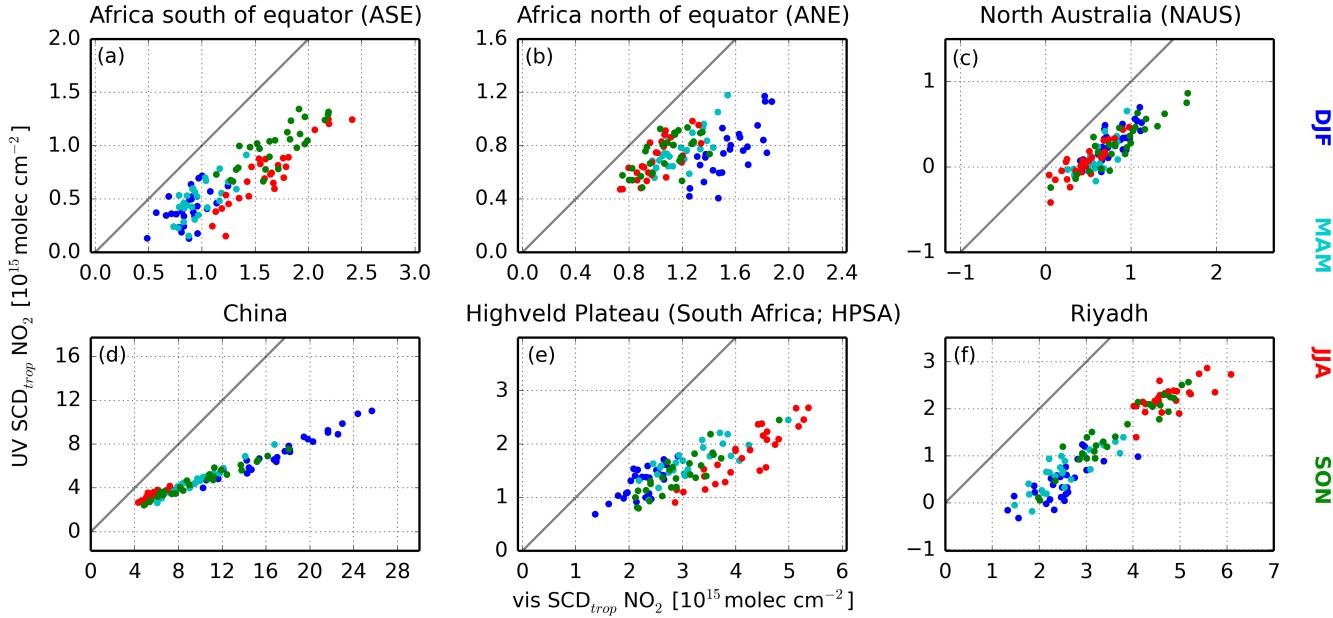

**Figure 10.** Scatter plot of tropospheric UV $NO_2$ SCDs against tropospheric vis $NO_2$ SCDs for the period $2007-2015$ for the six selected regions. Note the different axes. The seasons are color coded. The related correlation, slope and intercept of the regression line can be found in Tab. 3 and 4. December, January, February: DJF; March, April, May: MAM; June, July, August: JJA; September, October, November: SON.



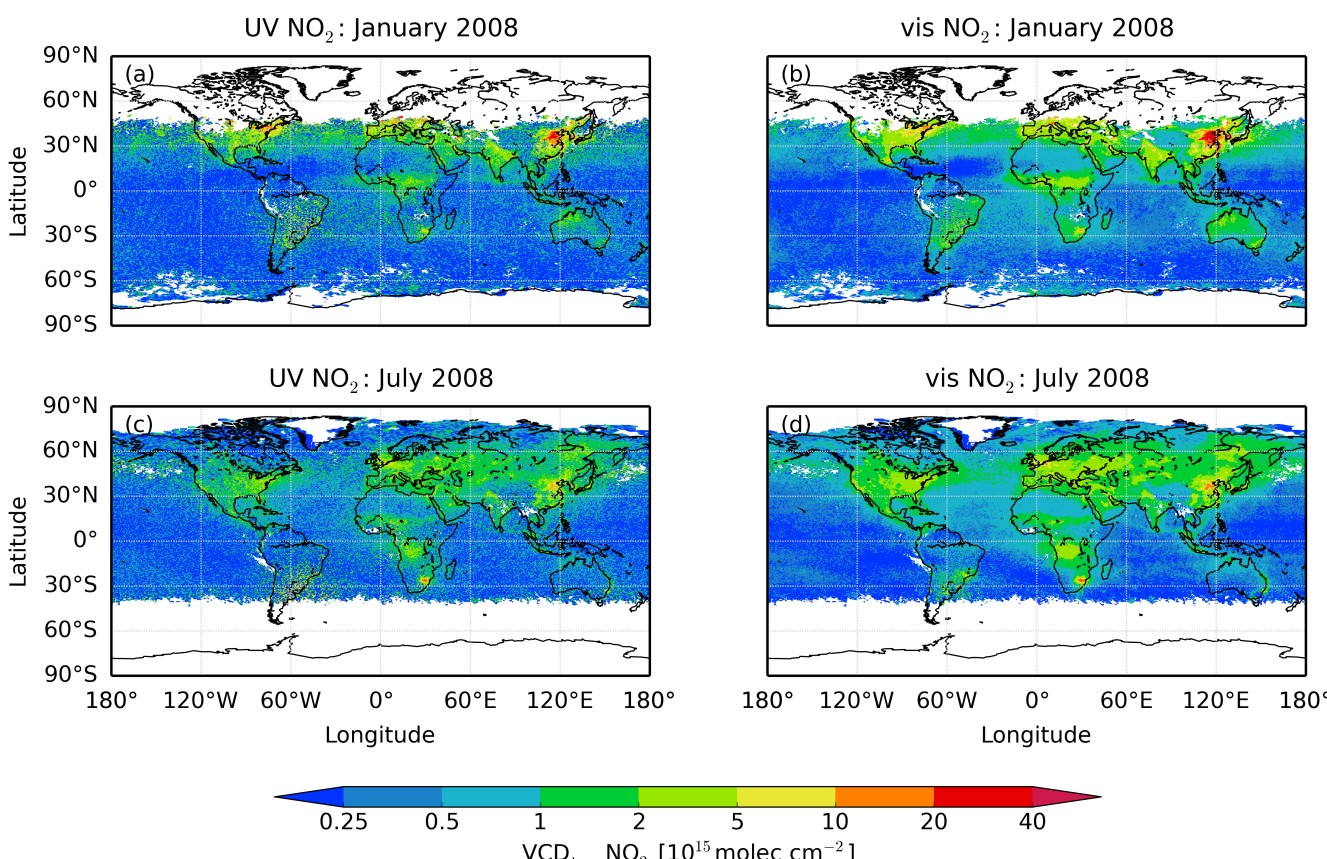

**Figure 11.** Monthly mean tropospheric NO₂ VCDs for (a, b) January and (c, d) July 2008. (a, c) UV spectral range and (b, d) vis spectral range.





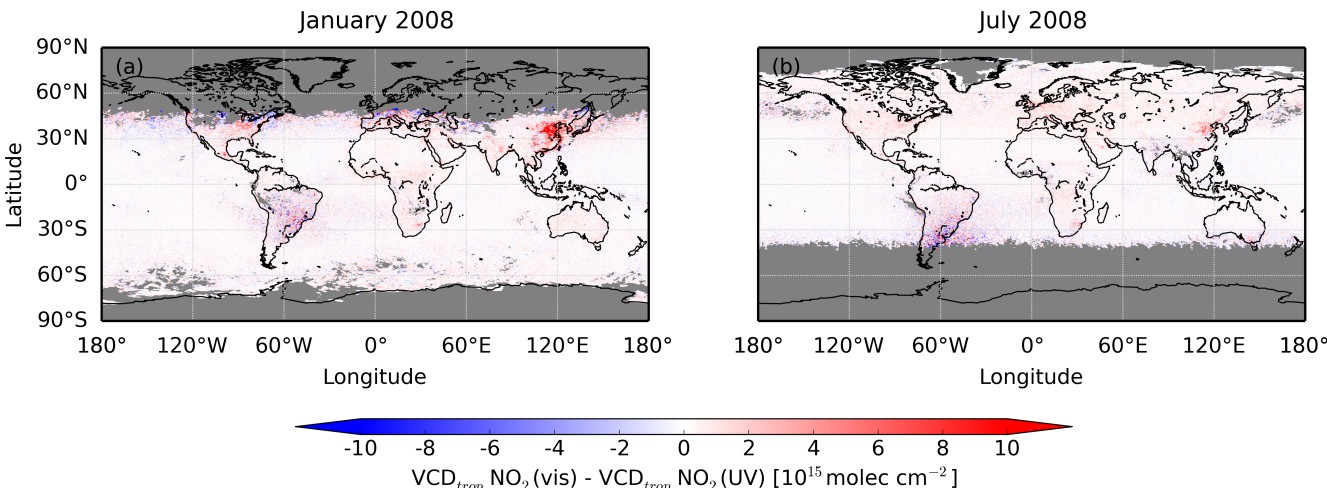

**Figure 12.** Absolute difference between monthly mean tropospheric NO₂ VCDs in the UV and vis spectral range. Differences for (a) January and (b) July 2008. Dark gray shaded area: no NO₂ values available.

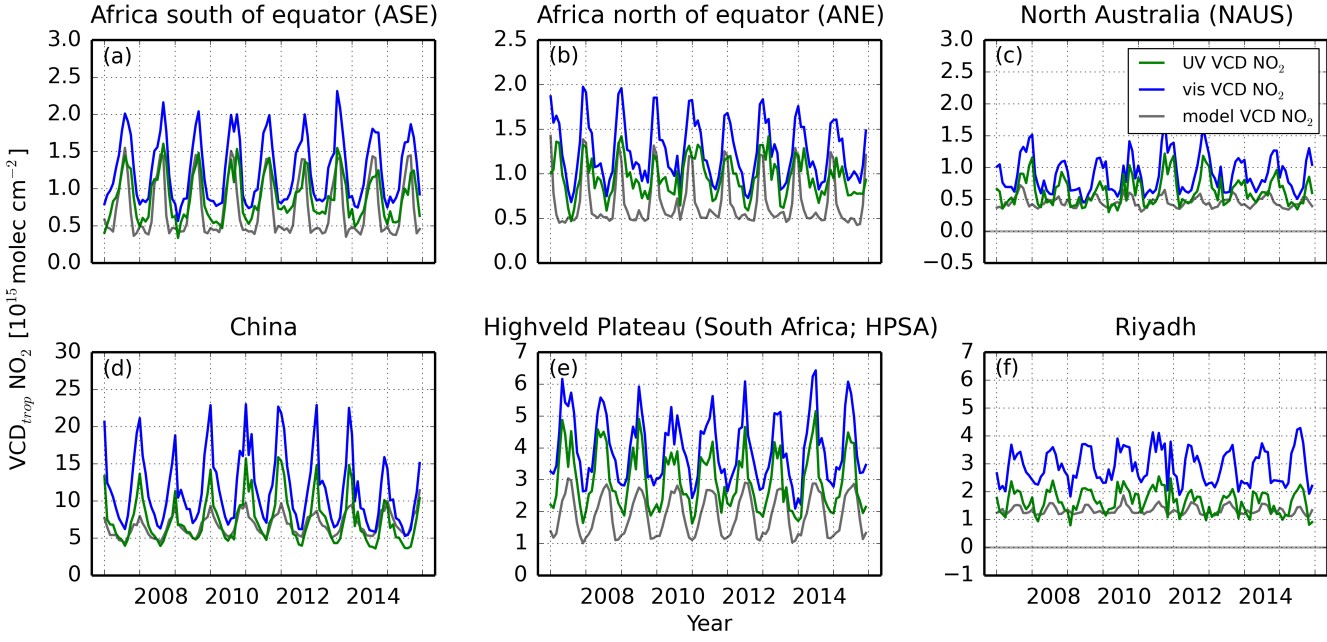

**Figure 13.** Time series 2007−2015 of tropospheric NO₂ VCDs for the UV (green line) and vis (blue line) spectral range as well as the TM5-model VCDs (gray line) for different regions. Note the different y-axes, the same as in Fig. 9. The vertical lines indicate January of the individual years. (a−c) biomass burning regions and (d−f) regions with anthropogenic air pollution.





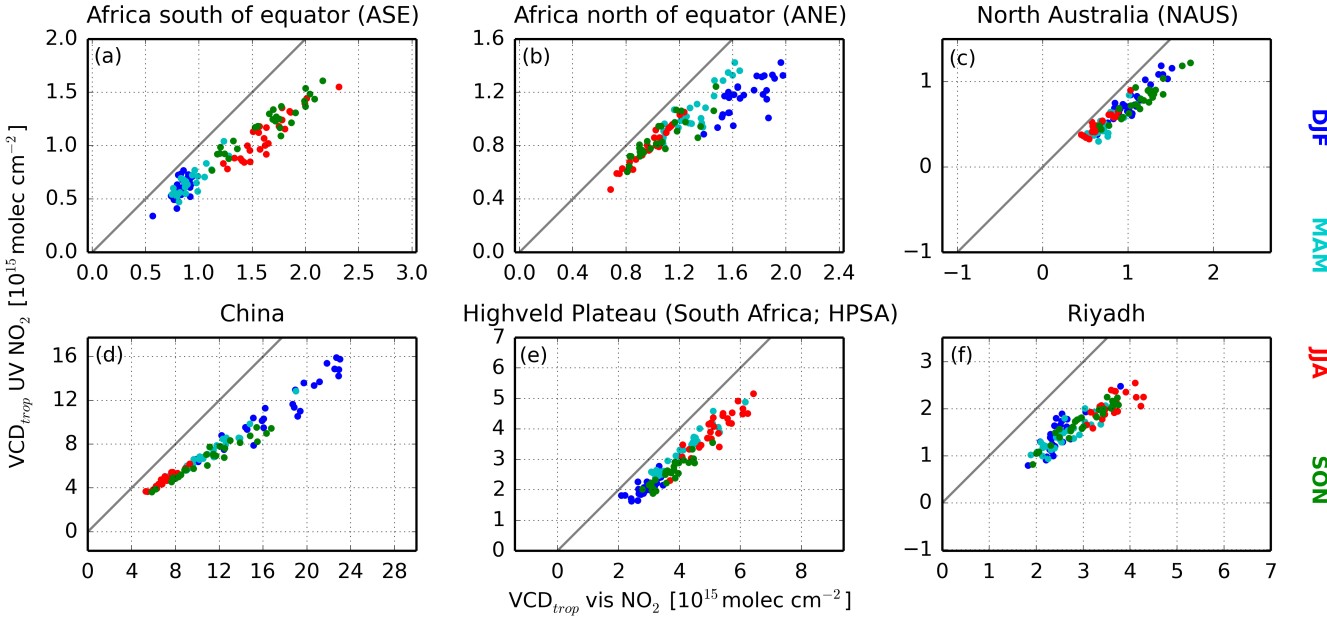

**Figure 14.** Scatter plot of tropospheric UV $NO_2$ VCDs against tropospheric vis $NO_2$ VCDs for the period 2007–2015 for the six selected regions. Note the different axes. The seasons are color coded. The related correlation, slope and intercept of the regression line can be found in Tab. 5 and 6. December, January, February: DJF; March, April, May: MAM; June, July, August: JJA; September, October, November: SON.




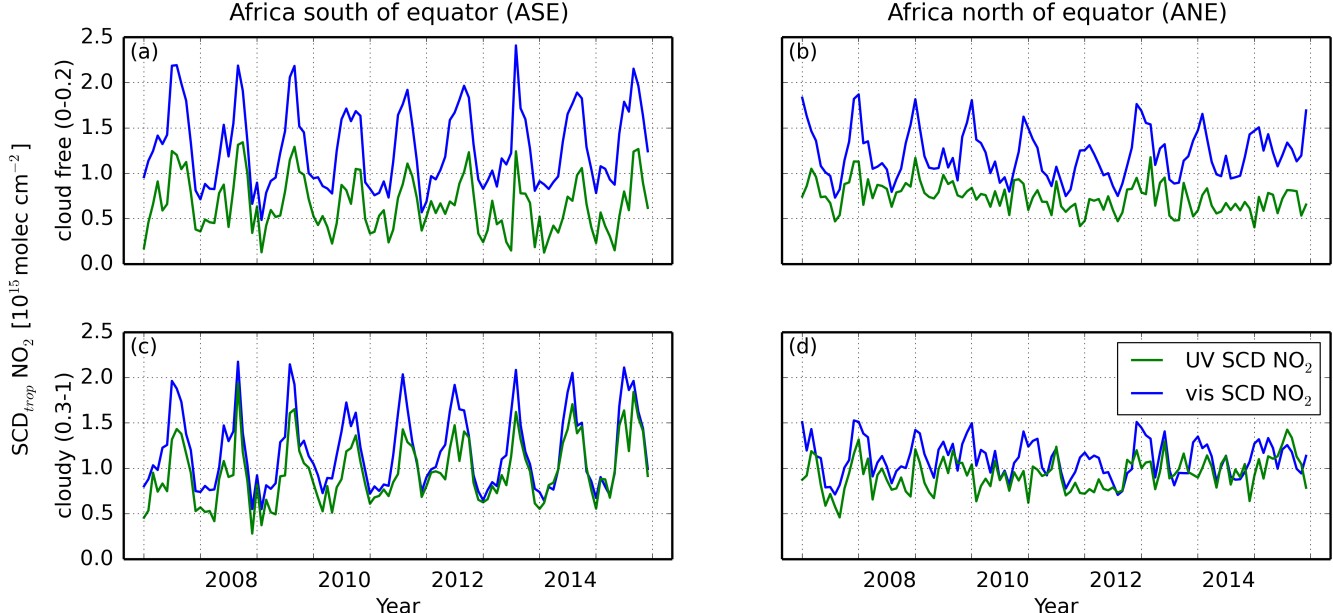

**Figure A1.** Tropospheric NO₂ time series for ASE (a, c) and ANE (b, d) for cloud free scenes ($\leq 20\,\%$ cloud fraction; a, b) and for cloudy scenes ($\geq 30\,\%$ cloud fraction; c, d). The vertical lines indicate January of the individual years. (a, b) are identical to Fig. 9 a, b and are replicated here to facilitate direct comparison.

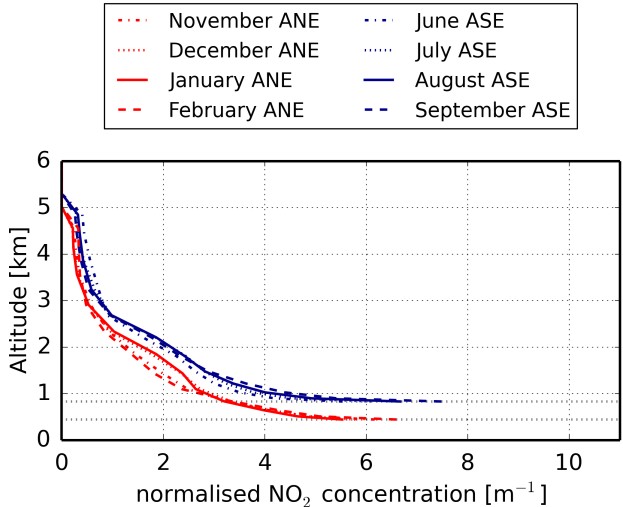

**Figure A2.** Average profiles of normalised NO₂ concentration for the three regions ANE and ASE for 2008, derived from the TM5-model for 09:00 LT. The gray dashed lines indicating the surface height of the two regions.





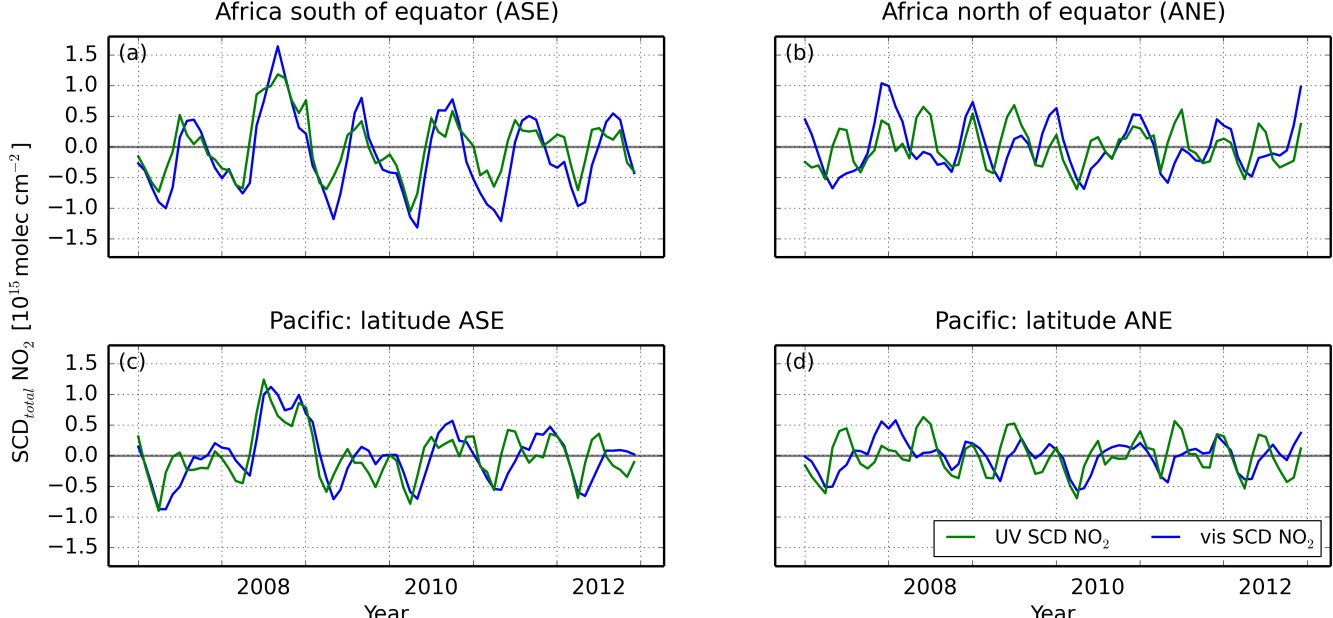

**Figure A3.** Time series of detrended total NO₂ SCDs for the UV and vis spectral range for ASE (a) and ANE (b) as well as detrended time series over the Pacific reference sector area for the latitude of ASE (c) and ANE (d). The vertical lines indicate January of the individual years.

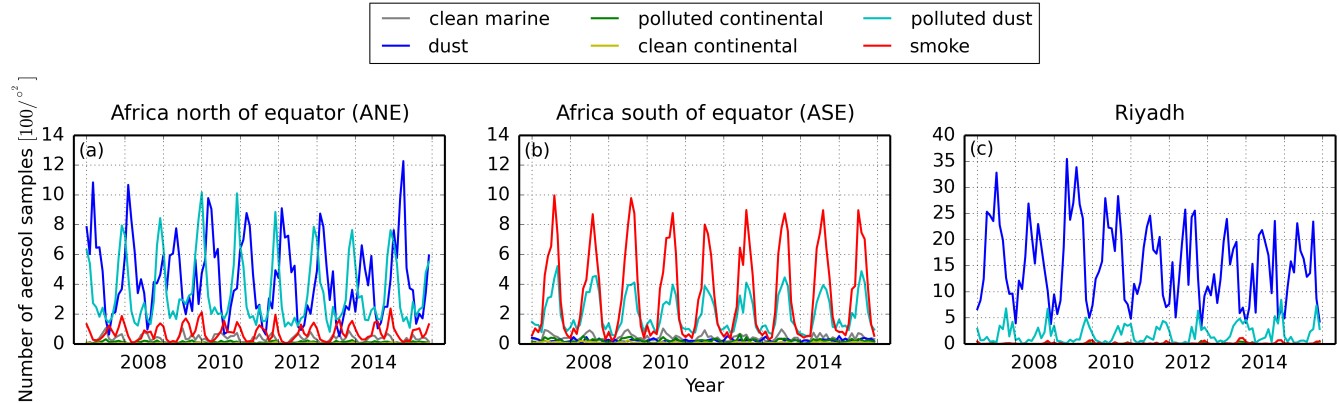

**Figure A4.** Time series for the number of samples of aerosol-types for three different regions (CALIPSO data). To calculate the time series for the individual areas, the amount of aerosol in the grid box was weighted by the contribution of the grid box to the region. Note the different y-axes. The vertical lines indicate January of the individual years.



**Table 1.** Fit settings for the NO$_2$ retrievals in the UV and the vis spectral range.

|                      | UV spectral range | vis spectral range |
| -------------------- | ----------------- | ------------------ |
| fitting window       | 342−361.5 nm      | 425−450 nm         |
| polynomial degree    | 4                 | 4                  |
| cross sections       | O$_3$, NO$_2$, O$_4$, BrO, HCHO, Ring | NO$_2$, O$_3$, O$_4$, H$_2$O, Ring |
| instrumental function| Zeta              | −                  |

**Table 2.** Selected regions for NO$_2$ time series with their abbreviation and their location.

| Region | Abbreviation | geographical location |
| ------ | ------------ | --------------------- |
| Africa south of the equator | ASE | 5° S − 20° S and 10° E − 40° E |
| Africa north of the equator | ANE | 0° N − 10° N and 15° W − 40° E |
| North Australia | NAUS | 15° S − 25° S and 120° E − 145° E |
| China | — | 30° N − 40° N and 110° W − 123° E |
| Highveld Plateau in South Africa | HPSA | 22° S − 30° S and 27° E − 34° E |
| Riyadh | — | 23° N − 26° N and 45° E − 48° E |

**Table 3.** Correlation between UV NO$_2$ SCDs and vis NO$_2$ SCDs for the six selected regions.

| Region | Annual | DJF | MAM | JJA | SON |
| ------ | ------ | --- | --- | --- | --- |
| ASE | 0.87 | 0.50 | 0.69 | 0.92 | 0.80 |
| ANE | 0.53 | 0.75 | 0.77 | 0.76 | 0.63 |
| NAUS | 0.87 | 0.76 | 0.75 | 0.87 | 0.90 |
| China | 0.98 | 0.98 | 0.99 | 0.83 | 0.98 |
| HPSA | 0.82 | 0.74 | 0.80 | 0.90 | 0.87 |
| Riyadh | 0.95 | 0.73 | 0.86 | 0.70 | 0.95 |





**Table 4.** Slope and intercept between UV NO$_2$ SCDs and vis NO$_2$ SCDs for the six selected regions. Intercept in 1e15 molec cm$^{-2}$. In the brackets the standard errors for both values are shown.

| Region | Annual | DJF | MAM | JJA | SON |
|---|---|---|---|---|---|
| ASE | 0.66/ -0.19 | 0.98/ -0.45 | 0.96/ -0.44 | 0.83/ -0.61 | 0.65/ -0.10 |
| | (0.03/ 0.05) | (0.23/ 0.20) | (0.17/ 0.16) | (0.07/ 0.11) | (0.09/ 0.15) |
| ANE | 0.40/ 0.26 | 1.08/ -0.93 | 0.84/ -0.23 | 0.92/ -0.23 | 0.67/ 0.01 |
| | (0.05/ 0.06) | (0.16/ 0.26) | (0.12/ 0.15) | (0.14/ 0.14) | (0.13/ 0.14) |
| NAUS | 0.74/ -0.32 | 0.89/ -0.41 | 1.04/ -0.50 | 0.78/ -0.30 | 0.71/ -0.34 |
| | (0.04/ 0.03) | (0.13/ 0.11) | (0.16/ 0 09) | (0.08/ 0.04) | (0.06/ 0.06) |
| China | 0.37/ 1.04 | 0.42/ -0.07 | 0.46/ 0.25 | 0.49/ 0.52 | 0.37/ 0.89 |
| | (0.01/ 0.08) | (0.02/ 0.30) | (0.02/ 0.14) | (0.06/ 0.36) | (0.02/ 0.16) |
| HPSA | 0.41/ 0.24 | 0.74/ -0.37 | 0.46/ 0.21 | 0.68/ -1.03 | 0.60/ -0.32 |
| | (0.03/ 0.09) | (0.12/ 0.26) | (0.06/ 0.21) | (0.06/ 0.27) | (0.06/ 0.19) |
| Riyadh | 0.74/ -1.26 | 0.63/ -1.15 | 0.64/ -0.97 | 0.46/ 0.06 | 0.71/ -1.10 |
| | (0.02/ 0.08) | (0.10/ 0.26) | (0.07/ 0.19) | (0.08/ 0.40) | (0.04/ 0.17) |

**Table 5.** Correlation between UV NO$_2$ VCDs and vis NO$_2$ VCDs for the six selected regions.

| Region | Annual | DJF | MAM | JJA | SON |
|---|---|---|---|---|---|
| ASE | 0.96 | 0.67 | 0.88 | 0.92 | 0.94 |
| ANE | 0.91 | 0.74 | 0.91 | 0.98 | 0.86 |
| NAUS | 0.94 | 0.92 | 0.78 | 0.93 | 0.94 |
| China | 0.98 | 0.96 | 0.98 | 0.96 | 0.97 |
| HPSA | 0.96 | 0.78 | 0.97 | 0.90 | 0.92 |
| Riyadh | 0.91 | 0.82 | 0.87 | 0.67 | 0.96 |



**Table 6.** Slope and intercept between UV NO$_2$ VCDs and vis NO$_2$ VCDs for the six selected regions. Intercept in 1e15 molec cm$^{-2}$. In the brackets the standard errors for both values are shown.

| Region | Annual | DJF | MAM | JJA | SON |
|--------|--------|-----|-----|-----|-----|
| ASE | 0.70/ 0.01 | 1.51/ -0.65 | 0.83/ -0.11 | 0.85/ -0.29 | 0.67/ 0.08 |
| | (0.02/ 0.02) | (0.28/ 0.23) | (0.09/ 0.08) | (0.07/ 0.11) | (0.05/ 0.08) |
| ANE | 0.63/ 0.18 | 0.80/ -0.20 | 0.96/ -0.19 | 0.98/ -0.17 | 0.72/ 0.09 |
| | (0.03/ 0.04) | (0.13/ 0.21) | (0.08/ 0.11) | (0.04/ 0.04) | (0.08/ 0.09) |
| NAUS | 0.76/ -0.05 | 0.92/ -0.20 | 1.04/ -0.26 | 0.88/ -0.09 | 0.74/ -0.07 |
| | (0.03/ 0.02) | (0.08/ 0.08) | (0.15/ 0.11) | (0.07/ 0.04) | (0.05/ 0.06) |
| China | 0.64/ -0.04 | 0.69/ -0.78 | 0.67/ -0.25 | 0.67/ -0.05 | 0.54/ 0.65 |
| | (0.01/ 0.15) | (0.04/ 0.73) | (0.02/ 0.28) | (0.04/ 0.26) | (0.03/ 0.31) |
| HPSA | 0.86/ -0.47 | 0.69/ 0.07 | 0.86/ -0.25 | 0.92/ -0.80 | 0.78/ -0.34 |
| | (0.02/ 0.10) | (0.10/ 0.29) | (0.04/ 0.18) | (0.08/ 0.44) | (0.06/ 0.23) |
| Riyadh | 0.66/ -0.26 | 1.09/ -1.27 | 0.69/ -0.36 | 0.63/ -0.20 | 0.66/ -0.28 |
| | (0.03/ 0.08) | (0.14/ 0.34) | (0.07/ 0.19) | (0.11/ 0.41) | (0.04/ 0.12) |