# Peer review of "Investigation of NO2 vertical distribution using two DOAS retrievals for GOME-2A measurements in the UV and vis spectral range"

_Atmospheric Measurement Techniques, 2017_

## Referee Comment (RC1) · Anonymous Referee #2 · 15 Nov 2017

Behrens et al. present a study showing the relationship between UV and VIS tropospheric NO2 VCDs and how it can relate to information on vertical distribution of NO2 in the troposphere. Although the paper is well written, it appears to be rather deceiving. I find the paper too qualitative and the reader could expect an attempt to effectively derive some information on the vertical distribution of NO2 from the combination of UV and vis NO2 measurements. The authors provide a number of possible reasons for the differences between UV and vis NO2 VCDs (a-priori profiles, effect of clouds/aerosols, etc) that are all plausible (and speculative) but there is no clear way forward. They almost conclude on the current impossibility to derive profile information. A weak point is that it is difficult to separate possible errors in the retrievals (in the UV spectral fits)

[Figure]

from real effects.

To be published in AMT, the following points need to be addressed:

-a comprehensive error analysis on the UV retrievals needs to be undertaken.

-section 2.2: the effect of T on Uv+vis NO2 retrievals is not well discussed. NO2 cross sections are varying with T but only one T cross-section is included in the fits (both for UV and VIS DOAS fits). What is the impact on the results and conclusions of this study?

-section 2.6: what is the uncertainty due to errors in SSR?

-section 3.3, p14, l20: it is written that VCD differences are small but it is hard to judge as only absolute values for the differences are shown. It would be better to incorporate relative differences as well as proper error calculation (see comment above).

-p15: it is not clear how the CTM profiles should be changed to reconcile the vis, uv and modeled VCDs.

-p16, l25-26: the reading nearly suggests that it could be better to use UV retrievals but it is quite unlikely given the larger uncertainties.

-p18, l35: It is stated that 'concurrent measurement of the same air mass from different view geometries could yield insight on aerosols and vertical distribution' but these measurements do not exist.

Minor comments:

-p2, l34-35: I find misleading that the examples on past studies on vertical profiling are mostly unrelated to NO2 retrievals as given in the present paper (which considers optically thin atmosphere as for NO2).

-p3, l13: ..path within the NO2 layer relative to the vertical path.

-p3, l15: environmental effects is too vague.
-throughout the manuscript, the author often use the word 'visibility' to express the idea that NO2 signal is more clear 'visible' in some spectral range. It is ambiguous as it might be interpreted by 'visible wavelength'.

-p3: the example of bAMFs should be presented here.

-p3: the effect of BRF on NO2 retrievals is not developed enough as possible reason between UV and vis

-p3, l34: sofar –> so far.

-p5, l 15: "strong absorption lines" -> "strong differential absorption lines"

-section 2.5: for the SCDs stratospheric correction, are the averages performed for both UV and VIS data separately?

-p10, l16-17: this is a bit contradictory. If it is below the detection, then how meaningful is the 0.6 SCD ratio?

-p13, l8, first word: India->China?

-p13, l18: SZA is lower-> SZA is higher?

---

## Referee Comment (RC2) · Anonymous Referee #1 · 20 Dec 2017

The paper "Investigation of NO2 vertical distribution using two DOAS retrievals for GOME-2A measurements in the UV and vis spectral range" by Behrens et al. presents a retrieval of NO2 vertical columns in the UV spectral range and discusses how far the comparison to the "standard" retrieval in the blue spectral range provides information on the vertical distribution of NO2 within the troposphere. The manuscript is well written, clearly structured and overall convincing in its conclusions. It matches well in the scope of AMT.

I have two fundamental comments:

1. The goal of investigating the vertical distribution of NO2, as stated in the title, seems

quite ambitious to me; one measured spectrum just provides information on the integrated column (SCD), without any information on the vertical distribution. It is not clear to me why the authors seem to have expected to get direct quantitative information on the vertical distribution by just adding one further piece of information (the SCD at a different wavelength). So the aim of deriving vertical information in the troposphere seems to be rather a second step; as first step, I would have expected improved stratospheric estimates. The separation of stratospheric and tropospheric column is still a fundamental challenge for the retrieval of tropospheric NO2 VCDs. While it is often no problem over highly polluted regions, it is still a crucial prerequisite for accurate emission estimates of large-scale sources such as soil emissions. Thus I miss some discussion about how far a UV retrieval might improve stratospheric estimates. Empirical approaches such as the RSM and modifications (allowing for longitudinal gradients) as well as assimilation approaches use the measured total column for the estimation of the stratospheric fraction. This requires some kind of a-priori knowledge on how large the tropospheric contribution is expected to be. For a UV retrieval, the sensitivity to the tropospheric column is weaker. Thus, a modified RSM approach might benefit from UV measurements since it can include more measurements over weakly polluted regions. This topic should be discussed in the manuscript.

2. The authors use the established blue spectral range plus a fit window in the UV. But what about the green spectral range? As shown in Fig. 1, the NO2 cross section still shows absorption bands above 550 nm. This would be another complementary piece of information, even more sensitive towards the lower troposphere than the standard retrieval. Of course the noise will be higher, and the choice of the fitting window and the water vapour cross section might be challenging. I understand that such a "green" retrieval would require major data processing; but still the authors should discuss the potential of adding additional wavelengths for the goal of assessing profile information, and I would appreciate if they could provide some sensitivity studies.

Minor comments:

Page 3 line 32: "can provide information about the accuracy": I would suggest a different formulation such as "allows to infere the validity of a-priori profiles".

Page 4 line 12: should be "summary and conclusions".

Section 2.1: Please add Munro et al., 2016 also as reference for GOME-2 A.

Page 4 line 30: add "before launch" or similar.

Section 2.2: The details of the cross sections and respective references might be shifted to Table 1.

Page 5 line 6: Which were the criteria for identifying "best results"?

Page 5 line 11: For a focus on stratospheric patterns, this would probably be crucial.

Page 5 line 12: So daily Earth is the alternative, but what is the default?

Page 5 line 20: Details and references for $O_4$ (vis) and $H_2O$ are missing (might also be added to Table 1).

Page 6 line 21: What does "upper atmospheric" mean?

Page 6 line 30: What is meant by "this correction method"? The RSM? Then write it. Or any correction of the RSM close to the polar vortex? Then provide further details.

Page 7 line 2: "This correction" -> RSM?

Page 7 line 5: "no negative values are expected": if the model is correct!

Page 7 line 9: "... using a linear approach" -> "the $NO_2$ VCDs are scaled by a correction factor linear in T in order to correct for the temperature dependency of the $NO_2$ cross section, as suggested in Boersma..."

Page 12 line 24: avoid "believe"; you have provided several arguments for this conclusion.

---

## Author Comment (AC2) · 6 Mar 2018

**Author reply to Referee #2**

Lisa K. Behrens et al.

March 6, 2018

We thank Referee #2 for carefully reading our manuscript and for the helpful comments which will improve the quality of our manuscript. We will reply to the comments point by point.
* * *
Legend:
- referee comments
- authors comments
- **changed text in the manuscript**
* * *
Behrens et al. present a study showing the relationship between UV and VIS tropospheric NO2 VCDs and how it can relate to information on vertical distribution of NO2 in the troposphere. Although the paper is well written, it appears to be rather deceiving. I find the paper too qualitative and the reader could expect an attempt to effectively derive some information on the vertical distribution of NO2 from the combination of UV and vis NO2 measurements. The authors provide a number of possible reasons for the differences between UV and vis NO2 VCDs (a-priori profiles, effect of clouds/aerosols, etc) that are all plausible (and speculative) but there is no clear way forward. They almost conclude on the current impossibility to derive profile information. A weak point is that it is difficult to separate possible errors in the retrievals (in the UV spectral fits) from real effects.

We agree with the referee that the title may suggest that combined UV and vis $NO_2$ measurements would be provided information about the $NO_2$ profiles in this manuscript which may lead to confusion. Therefore, we will change the title to: "GOME-2A retrievals of tropospheric $NO_2$ in different spectral ranges - influence of penetration depth". Furthermore, we will add maps with retrieved top-altitudes of $NO_2$ layer height (Fig. 1) in the revised manuscript in Sect. 3.1. For the altitude retrieval simple box profiles are assumed for tropospheric $NO_2$. A seasonal dependency of the retrieved altitude can be clearly observed in the global maps.

[Figure]

Figure 1: Monthly mean top-altitudes retrieved from the ratio between the UV and blue spectral range. To retrieve the altitude, box profiles are assumed for the tropospheric $NO_2$. The light grey coloured values indicate values which are below the threshold defined for the ratio (see Fig. 8 manuscript; will be added in a revised version)

Furthermore, we agree that it is difficult to separate possible errors in the retrieval from real effects. With the help of the referee's comments we will make the manuscript more quantitative, by including an error discussion and the discussion of the temperature dependency of SCDs.

**To be published in AMT, the following points need to be addressed:**

-a comprehensive error analysis on the UV retrievals needs to be undertaken.

Done — we will include the following table in Sect. 2.7 (2.8 in the revised manuscript):

Table 1: Errors for the UV and vis spectral range.

| | errors UV | errors vis |
|---|---|---|
| total SCDs | $1.8 \times 10^{15}\,\mathrm{molec\,cm^{-2}}$ | $0.6 \times 10^{15}\,\mathrm{molec\,cm^{-2}}$ |
| | (calculated above the Pacific Ocean, see Fig. 4 (manuscript)) | |
| strato. VCDs | $7.4 \times 10^{14}\,\mathrm{molec\,cm^{-2}}$ | $2.1 \times 10^{14}\,\mathrm{molec\,cm^{-2}}$ |
| | (see Fig. 4 (manuscript)) | |
| SSR | 40% at 320 nm | 5% at 500 nm |
| | (Kleipool et al., 2008) | (Kleipool et al., 2008) |
| 0.05, increase 0.01 | BAMF increases 9% | BAMF increases 11% |
| | (338 nm, Lorente et al., 2017) | (440 nm, Lorente et al., 2017) |
| AMF | | |
|   cloud fraction | $0 - 30\%$, Boersma et al., 2004 | |
|   cloud height | $< 10\%$, Boersma et al., 2004 | |
|   aerosols | not included in our calculations $\rightarrow$ 15%, Boersma et al., 2004 | |
|   profile shape | $< 15\%$ (regions with little $NO_2$: $> 50\%$), Boersma et al., 2004 | |

-section 2.2: the effect of T on Uv+vis NO2 retrievals is not well discussed. NO2 cross sections are varying with T but only one T cross-section is included in the fits (both for UV and VIS DOAS fits). What is the impact on the results and conclusions of this study?

The $NO_2$ cross sections have a temperature dependency, which differs for the UV and vis spectral range as shown in Fig. 2 and Fig. 3 exemplarily for the two spectral ranges. This temperature dependency can further increase the differences between the two spectral ranges. Figure 4 and Fig. 5 show the altitude dependent sensitivity of the scaling coefficients for the UV and vis spectral ranges. The temperature dependency influences the tropospheric as well as the stratospheric $NO_2$ measurements. For the stratospheric $NO_2$ measurements, the sensitivity of the UV $NO_2$ spectral range is approximately 10% higher than for the vis spectral range. Close to the surface, the difference in temperature sensitivity is up to 10% stronger in the vis spectral

range. For VCDs, the temperature dependency of the $NO_2$ cross section is scaled by a linear correction factor for both spectral ranges, as suggested in Boersma et al., 2004. In the revised manuscript, we will include this issue in our discussion, but we will move this point to Sect. 2.4 (p. 5, l. 27, manuscript):

[...] **Furthermore, the measurement sensitivity for $NO_2$ decreases towards the surface. This can be clearly observed in the BAMF (Fig. 2, manuscript). This effect is enhanced by the temperature dependency of the $NO_2$ cross section. The temperature dependency influences the tropospheric as well as the stratospheric $NO_2$ measurements (see Fig. 4 and Fig. 5). For the stratospheric $NO_2$ retrieval, the sensitivity in the UV spectral range is up to 10% higher than for the vis spectral range. Close to the surface, the temperature sensitivity is up to 10% stronger in the vis spectral range compared to the UV spectral range. The temperature vertical sensitivity introduces a seasonal and a latitudinal dependency. This effect is stronger in the tropics than for higher / lower latitudes and in the mid-latitudes it is more pronounced in summer and less in winter.**

[**In combination with ...**]

[Figure]

Figure 2: NO$_2$ cross section and their temperature dependency for the UV spectral range. The scaling coefficient of the temperature dependency is calculated. (will be added in a revised manuscript, supplement)

[Figure]

Figure 3: NO$_2$ cross section and their temperature dependency for the vis spectral range. The scaling coefficient of the temperature dependency is calculated. (will be added in a revised manuscript, supplement)

[Figure]

Figure 4: Altitude dependency of the NO$_2$ scaling coefficient for China. The profiles are calculated for model data simulated with the TM5 model for 2008. (will be added in a revised manuscript, supplement)

[Figure]

Figure 5: Altitude dependency of the $NO_2$ scaling coefficient for ASE. The profiles are calculated for model data simulated with the TM5 model for 2008. (will be added in a revised manuscript, supplement)

-section 2.6: what is the uncertainty due to errors in SSR?

This issue has been addressed in the new Tab. 1.

-section 3.3, p14, l20: it is written that VCD differences are small but it is hard to judge as only absolute values for the differences are shown. It would be better to incorporate relative differences as well as proper error calculation (see comment above).

In response to the suggestion of the reviewer, we will show the relative differences in the revised manuscript (see Fig. 6 and Fig. 7). The figures with the absolute differences will be moved to the supplement.

[Figure]

Figure 6: Relative difference between monthly mean tropospheric $NO_2$ SCDs in the vis and UV spectral range. Differences for (a) January and (b) July 2008. Dark grey shaded area: no $NO_2$ values available. Light grey coloured values indicate values where the vis $NO_2$ is close to zero, which have been filtered out.

[Figure]

Figure 7: Relative difference between monthly mean tropospheric $NO_2$ VCDs in the UV and vis spectral range. Differences for (a) January and (b) July 2008. Dark grey shaded area: no $NO_2$ values available. Light grey coloured values indicate values which are filter out. The same filter as for Fig. 6 is used.

[Figure]

-p15: it is not clear how the CTM profiles should be changed to reconcile the vis, uv and modeled VCDs.

Figure 8: SCD, VCD, and AMF for different $NO_2$ profiles, calculated for one scenario in January over China with a SZA of 66°. Blue: true profile, the SCDs are calculated for this profile. Other color: changed input profiles for AMF calculation leading to changes in the retrieved VCDs.

As shown in Fig. 8, the profile shape has an influence on the retrieved VCDs, if the assumed and true profile shape do not agree. Exemplarily for one scenario over China in January with a SZA of 66°, the VCDs are calculated. To avoid possible measurement errors in the retrieved SCDs, the SCDs are also calculated with the radiative transfer model SCIATRAN. The "true" $NO_2$ profile is the blue curve in Fig. 8 a. Additionally, the $NO_2$ profiles were changed (colour coded) and AMF (Fig. 8 b) were calculated for these changed profiles which introduce differences in the retrieved VCDs (Fig. 8 d). The two profiles with lower $NO_2$ values than the true $NO_2$ profile close to the surface and with a slightly higher PBL and a smoother decrease of $NO_2$ values (red and green) for lead to a similar situation as observed in our study. The observed VCDs are higher for both spectral ranges than model VCDs and the VCDs for the UV spectral range are lower than for the vis spectral range. For a scenario with a much higher PBL, a constant mixing in the PBL (yellow), and a sharp decrease above, the model values are higher than the retrieved values which we have not observed in our study. For lower PBL with higher $NO_2$ values (cyan), the differences between the spectral ranges and the model VCDs are less pronounced. Therefore, our observations suggest that compared to the real $NO_2$ profiles, in the TM5

model the $NO_2$ is higher in the atmosphere with lower surface concentration values.
We will include Fig. 8 in the supplement and discuss this issue in more detail in the revised manuscript.

-p16, l25-26: the reading nearly suggests that it could be better to use UV retrievals but it is quite unlikely given the larger uncertainties.

We will change the sentence as follows:
**For ASE and ANE during biomass burning season, the UV $NO_2$ VCDs and the model VCDs agree quite well, while the vis $NO_2$ values are higher. It should however be kept in mind that in the vis retrieval, both the uncertainties and the contribution of the a priori are lower, making these results more reliable.**

-p18, l35: It is stated that 'concurrent measurement of the same air mass from different view geometries could yield insight on aerosols and vertical distribution' but these measurements do not exist.

We gave an example for developments for possible space-borne measurements in the future. Furthermore, there are already concurrent measurements of the same air mass. Therefore, there might be already the possibility to retrieve these kind of information from GOME-2 measurements using the instruments on the MetOp-A and MetOp-B. For example on the 01 January 2013 (before reduction of spatial resolution of GOME-2A) for a pixel of South Korea, the GOME-2A instrument has an overpass time of 2:04 UTC with a SZA of 61.86°, a LOS of -43.26°, and a RAA of -46.66°, whereas GOME-2B has an overpass time of 1:17 with a SZA of 66.84°, a LOS of 18.41°, and a RAA 137.35°.
We will change the sentence as follows:
**Future developments in space-based observation of the Earth's atmosphere could however increase the capability to retrieve vertical $NO_2$ distribution. For example, concurrent measurements of the same air mass from different viewing geometries could yield insight on aerosol types and vertical profiles which might be also possible for GOME-2A and GOME-2B observations.**

**Minor comments:**

-p2, l34-35: I find misleading that the examples on past studies on vertical profiling are mostly unrelated to NO2 retrievals as given in the present paper (which considers optically thin atmosphere as for NO2).

The reviewer is right that we gave examples which are unrelated to $NO_2$. However, a similar method has so far not been used before for an optically thin gas. Therefore, we used examples for ozone which is an optically thick gas. We will point out this differences more clearly and change the paragraph as follows in the revised manuscript:

[...] **In consideration of this fact, knowledge of the vertical distribution of** $NO_2$ **can be gained by combining measurements at different wavelengths. The idea of using the penetration depth in the UV to determine vertical profiles of ozone was first proposed by Singer and Wentworth (1957). The use of the temperature dependence of the Huggins absorption bands coupled with penetration depth was proposed to retrieve information about the vertical profile of ozone in the troposphere (Chance et al., 1997). Here, will use a similar method for the optically thin trace gas** $NO_2$**.**

-p3, l13: ..path within the NO2 layer relative to the vertical path.

Done.

-p3, l15: environmental effects is too vague.

Done — we will change the sentence as follows in the revised manuscript:
**AMFs are calculated by radiative transfer models, which take into account the viewing geometry and environmental effects, e.g., scattering processes in the Earth's atmosphere, SSR and the vertical distribution of trace gases (Platt and Stutz, 2008).**

-throughout the manuscript, the author often use the word 'visibility' to express the idea that NO2 signal is more clear 'visible' in some spectral range. It is ambiguous as it might be interpreted by 'visible wavelength'.

In revised the manuscript, we will replace the word 'visible'. Then, it will be only used for the visible spectral range. However we think, it is not necessary to replace the word 'visibility'.

We will move Fig. 2 to the introduction as suggested and refer to this figure in the later chapters in the revised manuscript.

-p3: the effect of BRF on NO2 retrievals is not developed enough as possible reason between UV and vis

Done — we will change the paragraph as follows in the revised manuscript: **..... For wavelengths in the UV, the BAMF in layers close to the ground is considerably smaller than for the vis spectral range (Fig. 1, revised manuscript; Fig. 2 manuscript). This effect is even less pronounced for longer wavelengths. In general, BAMFs for longer wavelengths have a smaller dependency on altitude compared with BAMFs for shorter wavelengths (Burrows et al., 2011). The altitude of highest sensitivity further depends on the solar zenith angle (SZA). For increasing SZAs, the altitude of highest sensitivity moves upwards to the stratosphere. Furthermore, the surface spectral reflectance (SSR) depends on the wavelength, and therefore, the SCDs are influenced by the SSR (Burrows et al., 2011). Generally, for the UV and vis spectral range the SSR is quite low between 2 and 30% depending on the surface type except for snow or ice (Burrows et al., 2011). For these kind of surface types, the SSR is lower in the UV than in the vis spectral range. For smaller SSR, the UV shows a stronger decrease and for larger SSR (e.g. snow) the UV shows stronger increase towards the surface thus the UV SCDs decrease or increase compared to the vis SCDs. Therefore, the SSR can strengthen the effect of the Rayleigh scattering which can be further increased by higher SZAs. Additionally, aerosols influence the measurements and also the visibility of $NO_2$ is influenced by the presence of aerosols (Burrows et al., 2011). Depending on the type and the optical thickness of aerosols the influences on the measurement differs.**

Done.

-p5, l 15: "strong absorption lines" –> "strong differential absorption lines"

Changed as suggested.

-section 2.5: for the SCDs stratospheric correction, are the averages performed for both UV and VIS data separately?

Yes, they are calculated separately for both spectral ranges — we will change the sentence as follows:

**For the SCDs, we use the "reference sector method" (Richter and Burrows, 2002; Martin et al., 2002) for both spectral ranges separately, in which a monthly average of SCDs measured over a presumably clean area above the Pacific (180° E to 210° E) is subtracted from all measurements per latitude band.**

-p10, l16-17: this is a bit contradictory. If it is below the detection, then how meaningful is the 0.6 SCD ratio?

Yes, that is true — we will change the sentence as follows:

**Finally, Fig. 8 (manuscript) shows SCD ratios over the well known shipping lane leading from South India to the Strait of Malacca.**

No, the paper from Hilboll et al. (2017) is about air pollution in India.

Yes — Done.

Additionally as suggested by Referee #1, we will change two main points in the revised manuscript:

1. We will add discussion about stratospheric $NO_2$ and show that it is so far not possible to improve the stratospheric $NO_2$ retrieval by using different wavelength ranges.

2. We will discuss the possibility of an additional fitting window in the green spectral range. The fitting window in the green spectral range

has a higher sensitivity to the lower troposphere. However, in the green spectral range interferences with the surface are clearly visible and therefore in the revised manuscript, we will include only a case study for China.

**References**

Boersma, K. F., Eskes, H. J., and Brinksma, E.: Error analysis for tropospheric NO2 retrieval from space, Journal of Geophysical Research, 109, D04 311, doi:10.1029/2003JD003962, 2004.

Burrows, J. P., Platt, U., and Borrell, P., eds.: The Remote Sensing of Tropospheric Composition from Space, Physics of Earth and Space Environments, Springer-Verlag Berlin Heidelberg, doi:10.1007/978-3-642-14791-3, 2011.

Chance, K., Burrows, J., Perner, D., and Schneider, W.: Satellite measurements of atmospheric ozone profiles, including tropospheric ozone, from ultraviolet/visible measurements in the nadir geometry: a potential method to retrieve tropospheric ozone, Journal of Quantitative Spectroscopy and Radiative Transfer, 57, 467–476, doi:10.1016/S0022-4073(96)00157-4, 1997.

Hilboll, A., Richter, A., and Burrows, J. P.: NO2 pollution over India observed from space - the impact of rapid economic growth, and a recent decline, Atmospheric Chemistry and Physics Discussions, 20, 1–18, doi: 10.5194/acp-2017-101, 2017.

Kleipool, Q. L., Dobber, M. R., de Haan, J. F., and Levelt, P. F.: Earth surface reflectance climatology from 3 years of OMI data, Journal of Geophysical Research Atmospheres, 113, doi:10.1029/2008JD010290, 2008.

Lorente, A., Folkert Boersma, K., Yu, H., Dörner, S., Hilboll, A., Richter, A., Liu, M., Lamsal, L. N., Barkley, M., De Smedt, I., Van Roozendael, M., Wang, Y., Wagner, T., Beirle, S., Lin, J. T., Krotkov, N., Stammes, P., Wang, P., Eskes, H. J., and Krol, M.: Structural uncertainty in air mass factor calculation for NO2and HCHO satellite retrievals, Atmospheric Measurement Techniques, 10, 759–782, doi:10.5194/amt-10-759-2017, 2017.

Martin, R. V., Chance, K., Jacob, D. J., Kurosu, T. P., Spurr, R. J. D., Bucsela, E., Gleason, J. F., Palmer, P. I., Bey, I., Fiore, A. M., Li, Q., Yantosca, R. M., and Koelemeijer, R. B. a.: An improved retrieval of tropospheric nitrogen dioxide from GOME, Journal of Geophysical Research, 107(D20), 4437, doi:10.1029/2001JD001027, 2002.

Richter, A. and Burrows, J. P.: Tropospheric NO2 from GOME measurements, Advances in Space Research, 29, 1673–1683, doi:10.1016/S0273-1177(02)00100-X, 2002.

---

## Author Response (AR1)

**Revised manuscript**

Dear Dr. Michel Van Roozendael,

we would like to take the opportunity to thank you for your efforts and that you accepted the editorship of our manuscript "GOME-2A retrievals of tropospheric  $NO_2$  in different spectral ranges - influence of penetration depth". Furthermore, we would like to thank you for the extension of the review period for the manuscript.

Please find enclosed a revised version of our manuscript where we implemented all comments by the referees. We revised the original manuscript according to their suggestions and provided also additional information as requested by the referees. In particular, we added an additional fitting window for the green spectral range and a section about stratospheric NO2. More details are provided on both the retrieval errors and temperature dependency of the NO2 cross section. Supporting Figures have been compiled into an extensive supplement to our manuscript. Furthermore, we changed again the title of section from "2.6 Removing stratospheric NO2" to "2.6 Conversion to tropospheric NO2 columns".

Below the author comments are provided that have already been uploaded to the AMT web page on 06 March 2018. We also provide here a version of the revised manuscript in which changes in comparison to the initial version are highlighted. We hope that with the submission of the author comments and the revision of the manuscript, our article will be accepted for publication in AMT.

Yours sincerely, Lisa Behrens (on behalf of the co-authors)

List of Attachments

- Author comments to Referee #1
- Author comments to Referee #2
- Revised manuscript with highlighted changes
- Revised manuscript
- Supplementary information

**Author reply to Referee #1**

Lisa K. Behrens et al.

March 6, 2018

We thank Referee #1 for carefully reading our manuscript and for the helpful comments which will improve the quality of our manuscript. We will reply to the comments point by point.

Legend:

- referee comments

- authors comments

- changed text in the manuscript

The paper "Investigation of NO2 vertical distribution using two DOAS retrievals for GOME-2A measurements in the UV and vis spectral range" by Behrens et al. presents a retrieval of NO2 vertical columns in the UV spectral range and discusses how far the comparison to the "standard" retrieval in the blue spectral range provides information on the vertical distribution of NO2 within the troposphere. The manuscript is well written, clearly structured and overall convincing in its conclusions. It matches well in the scope of AMT.

Thank you very much for the positive comments.

**I have two fundamental comments:**

1. The goal of investigating the vertical distribution of NO2, as stated in the title, seems quite ambitious to me; one measured spectrum just provides information on the integrated column (SCD), without any information on the vertical distribution. It is not clear to me why the authors seem to have expected to get direct quantitative information on the vertical distribution by just adding one further piece of information (the SCD at a different wavelength). ...

We agree with the referee that the investigation of the vertical distribution is ambitious and the title can be confusing, because we are not able to give a real profile shape. Therefore, we will change the title to: "GOME-2A retrievals of tropospheric NO2 in different spectral ranges - influence of penetration depth". Furthermore, we will add maps with retrieved top-altitudes of NO2 layer height (Fig. 1) in the revised manuscript in Sect. 3.1. For the altitude retrieval simple box profiles are assumed for tropospheric  $NO_2$ . A seasonal dependency of the retrieved altitude can be clearly observed in the global maps.

Figure 1: Monthly mean top-altitudes retrieved from the ratio between the UV and blue spectral range. To retrieve the altitude, box profiles are assumed for the tropospheric  $NO_2$ . The light grey coloured values indicate values which are below the threshold defined for the ratio (see Fig. 8 manuscript; will be added in a revised version)

... So the aim of deriving vertical information in the troposphere seems to be rather a second step; as first step, I would have expected improved stratospheric estimates. The separation of stratospheric and tropospheric column is still a fundamental challenge for the retrieval of tropospheric NO2 VCDs. While it is often no problem over highly polluted regions, it is still a crucial prerequisite for accurate emission estimates of large-scale sources such as soil emissions. Thus I miss some discussion about how far a UV retrieval might improve stratospheric estimates. Empirical approaches such as the RSM and modifications (allowing for longitudinal gradients) as well as assimilation approaches use the measured total column for the estimation of the stratospheric fraction. This requires some kind of a-priori knowledge on how large the tropospheric contribution is expected to be. For a UV retrieval, the sensitivity to the tropospheric column is weaker. Thus, a modified RSM approach might benefit from UV measurements since it can include more measurements over weakly polluted regions. This topic should be discussed in the manuscript.

We agree with the referee that the derivation of the vertical column is a second step and that it would be nice to improve the separation between troposphere and stratosphere which was indeed our original intention. Unfortunately, it was not possible to improve the stratospheric estimation with a UV NO2 retrieval as this suffers not only from noise but also from systematic biases. However as shown by the BAMFs (Fig. 6 (Fig. 2, manuscript, will be replaced in a revised version)), the UV spectral range has still a small sensitivity to the lower troposphere and therefore, the tropospheric contribution can also be observed in the UV spectral range. This tropospheric pollution can be observed in the total SCDs in both spectral ranges (Fig. 2, will be added in a revised version) as well as an additional fitting window in the green spectral region.

BAMFs show the largest differences between the UV and blue spectral range close to the ground ( $\sim$  factor 3). In higher altitudes of the troposphere, the differences are clearly reduced. As shown by Delon et al. (2008) and Stewart et al. (2008), also soil emissions (large-scale sources) are partly located in elevated layers which increases the visibility in the UV spectral range, and therefore, reduces the differences between the spectral ranges. A key issue here is the seasonal dependency between the spectral ranges (Fig. 3, Fig. 5; will be added in a revised version). In January, the  $vis_{blue}$  NO2 values are higher than the UV  $NO_2$  values especially above polluted areas (Fig. 3) whereas in July, both spectral ranges are similar or the UV  $NO_2$  values are higher than the visblue  $NO_2$  values. Similar offsets can be found between the blue and the green fitting window. Therefore by introducing an UV fitting window, improving the separation of stratosphere and troposphere is not possible and further investigations are needed for this special point. In our case, we are using tropospheric columns by subtracting the reference sector, and therefore, the differences are cancelled out to a large extent. We will add the following paragraph (between Sect. 2.4 and Sect. 2.5 in the old manuscript (2.6 in the new manuscript)):

**2.5 Stratospheric** $NO_2$**

Figure 5 (will be added in a revised version) shows the latitudinal and seasonal dependency for the three  $NO_2$  fitting windows. The seasonal dependency clearly differs between the three fitting windows also over regions dominated by stratospheric  $NO_2$ , especially for the green wavelength range strong interferences are observable. Although the differences are smaller between the UV and blue spectral range, they are clearly observable, for example at the equator (Figure 5 b; will be added in a revised version). In northern hemispheric summer the UV  $NO_2$  values are higher than the  $NO_2$  values derived from the blue spectral range whereas in northern hemispheric winter the  $NO_2$ values from the blue spectral range are slightly higher. Therefore, it is currently not possible to improve the stratospheric  $NO_2$  retrieval by using different wavelength ranges.